# Genome duplication in *Leishmania major* relies on persistent subtelomeric DNA replication

Jeziel Dener Damasceno[1]*, Catarina A Marques[1], Dario Beraldi[1], Kathryn Crouch[1], Craig Lapsley[1], Ricardo Obonaga[2], Luiz RO Tosi[2], Richard McCulloch[1]*

[1]The Wellcome Centre for Integrative Parasitology, University of Glasgow, Institute of Infection, Immunity and Inflammation, Glasgow, United Kingdom; [2]Department of Cell and Molecular Biology, Ribeirão Preto Medical School, University of São Paulo, Ribeirão Preto, Brazil

**Abstract** DNA replication is needed to duplicate a cell's genome in S phase and segregate it during cell division. Previous work in *Leishmania* detected DNA replication initiation at just a single region in each chromosome, an organisation predicted to be insufficient for complete genome duplication within S phase. Here, we show that acetylated histone H3 (AcH3), base J and a kinetochore factor co-localise in each chromosome at only a single locus, which corresponds with previously mapped DNA replication initiation regions and is demarcated by localised G/T skew and G4 patterns. In addition, we describe previously undetected subtelomeric DNA replication in G2/M and G1-phase-enriched cells. Finally, we show that subtelomeric DNA replication, unlike chromosome-internal DNA replication, is sensitive to hydroxyurea and dependent on 9-1-1 activity. These findings indicate that *Leishmania*'s genome duplication programme employs subtelomeric DNA replication initiation, possibly extending beyond S phase, to support predominantly chromosome-internal DNA replication initiation within S phase.

**\*For correspondence:**
jeziel.damasceno@glasgow.ac.uk (JDD);
Richard.McCulloch@glasgow.ac.uk (RMC)

**Competing interests:** The authors declare that no competing interests exist.

## Introduction

Once every cell cycle, a cell must completely duplicate its genome before segregating the two resulting copies into offspring cells. Genome duplication relies on DNA replication, a normally tightly controlled and high-fidelity reaction (*Burgers and Kunkel, 2017*). DNA replication is initiated at genomic loci termed origins, which are sequence-conserved features in prokaryotes (*Leonard and Méchali, 2013*). In contrast, with the exception of *Saccharomyces* and closely related yeasts (*Dhar et al., 2012*), replication origins in eukaryotes are not defined by conserved sequences. Instead, more elusive features, such as chromatin accessibility, transcription level and epigenetic elements (*MacAlpine et al., 2010*; *Cayrou et al., 2015*; *Deal et al., 2010*; *Dellino et al., 2013*; *Lombraña et al., 2013*; *Mesner et al., 2011*; *Chen et al., 2019*), are determinants of replication initiation activity. What is common to all known eukaryotic origins is that they are licensed through binding by the origin recognition complex (ORC), which recruits the replicative helicase, MCM2-7, during G1 (*Bleichert et al., 2017*). At the onset of S-phase origins are fired, initiating DNA synthesis that proceeds bi-directionally along the chromosomes.

Likely as a result of increasing genome size, DNA replication in eukaryotes is initiated at multiple origins per linear chromosome, with the number of origins being proportional to chromosome size (*Al Mamun et al., 2016*). To preserve genomic stability, origins licensed in G1 outnumber those that are fired in early S phase. Thus, in the event of failure of complete DNA synthesis from the fired origins, others can be activated to ensure complete genome duplication by completion of S phase

(*McIntosh and Blow, 2012*; *Alver et al., 2014*). However, unlike all previously characterised eukaryotes, mapping of DNA replication using Marker Frequency Analysis coupled with deep sequencing (MFA-seq) detected only a single clear region of replication initiation in each chromosome of Leishmania, a grouping of single-celled parasites (*Marques et al., 2015*). If these MFA-seq regions represent origins (as they do in *S. cerevisiae*) (*Müller and Nieduszynski, 2012*), these data would suggest a DNA replication programme in *Leishmania* that is unprecedented in eukaryotes and, indeed, contrasts with the multiple origins mapped in the chromosomes of *Trypanosoma brucei* (*Tiengwe et al., 2012a*), a kinetoplastid relative of *Leishmania* (see below). Moreover, this DNA replication programme is predicted to be insufficient to allow complete duplication of the larger *Leishmania* chromosomes during S phase (*Marques et al., 2015*), accounting for perhaps 50% of the chromosome complement, and is thus inadequate to secure complete genome duplication during prior to cell division. A further complication in the emerging understanding of *Leishmania* DNA replication is that a later study, which mapped short nascent DNA strands (SNS-seq) in asynchronous cells detected thousands of DNA synthesis initiation sites (hundreds per chromosome), therefore revealing a huge dichotomy with MFA-seq mapping (*Lombraña et al., 2016*). Indeed, 'DNA combing' analyses could detect DNA molecules with more than a single site of DNA synthesis (*Lombraña et al., 2016*; *Stanojcic et al., 2016*), though location within a chromosome could not be inferred and it could not be ruled out that extrachromosomal episomes, which arise frequently in *Leishmania* (*Ubeda et al., 2014*), were responsible for the DNA synthesis signals. These conflicting data raise questions, which we have sought to answer here, about the programme of DNA replication that *Leishmania* uses in order to effectively execute genome duplication.

*Leishmania* species are the causative agents of a spectrum of diseases, including skin-damage and fatal organ-failure leishmaniasis, affecting both humans and other animals worldwide (*Torres-Guerrero et al., 2017*). Leishmania belongs to the diverse kinetoplastid grouping (*Adl et al., 2019*; *Keeling and Burki, 2019*), which is evolutionarily distant from yeast, animals and plants, from where much of our understanding of eukaryotic DNA replication has emerged. Besides being of medical relevance (since several species and genera are parasites of insects and animals), kinetoplastids display unorthodoxies in core molecular processes. Notably, virtually all RNA polymerase (pol) II transcribed genes in Leishmania (*Martínez-Calvillo et al., 2004*) and *T. brucei* (*Kolev et al., 2010*) have been shown to be transcribed as multigene transcription units, each of which possesses a single poorly defined transcription start site that appears to be constitutively active (*Wedel et al., 2017*). The ubiquitous use of multigenic transcription is likely to be a universal feature of kinetoplastids (*Jackson et al., 2008*) and means that gene expression is perhaps exclusively regulated at the post-transcriptional level (*El-Sayed et al., 2005*; *Martínez-Calvillo et al., 2003*; *LeBowitz et al., 1993*; *Clayton, 2019*). Two further consequences arise from multigene transcription. First, one route for increased gene expression in the absence of transcriptional control is to increase gene copy number. In Leishmania this appears to be reflected in its remarkable genome plasticity (*Laffitte et al., 2016a*), manifested both as intra- and extra- chromosomal genome-wide gene copy-number variation (*Iantorno et al., 2017*; *Ubeda et al., 2008*) and as mosaic aneuploidy (*Rogers et al., 2011*; *Sterkers et al., 2011*; *Bussotti et al., 2018*; *Prieto Barja et al., 2017*). Second, in *T. brucei*, ORC and origin localisation is found at the boundaries of the multigene transcription units, where transcription initiates and/or terminates, perhaps limiting impediments to RNA pol movement (*Tiengwe et al., 2012a*). MFA-seq mapping also identified DNA replication initiation sites exclusively at transcription boundaries in Leishmania (*Marques et al., 2015*), whereas most SNS-seq mapped initiation sites are within the multigene transcription units, where they appear to coincide with trans-splicing and polyadenylation sites at which mature mRNAs are generated (*Lombraña et al., 2016*).

To attempt to clarify the current unclear picture of how DNA replication is executed in Leishmania, this study describes a modified MFA-seq strategy with which we demonstrate the existence of previously undetected sites of DNA replication initiation in the chromosomes of *L. major*. Consistent with previous MFA-seq mapping, DNA replication initiation in early S phase could only be clearly detected from a single internal region in each chromosome, which was marked by simultaneous accumulation of acetylated histone H3 (AcH3), a novel modified base, termed base J, and the putative kinetochore factor KKT1. In addition, we now describe DNA replication initiation that is proximal to the chromosome telomeres and only shows co-localisation with AcH3. DNA replication initiation from the subtelomeres is detected not only in early S phase cells, but also in cells enriched in late S, G2/M and G1 phases of the cell cycle. Furthermore, we show that telomere-proximal DNA

replication activity, unlike DNA replication from the core sites, is sensitive to replication stress and depends on RAD9 and HUS1, which are subunits of the 9-1-1 complex that acts in the replication stress response pathway (*Damasceno et al., 2013*; *Damasceno et al., 2016*; *Damasceno et al., 2018*). Thus, we reveal that two potentially distinct forms of DNA synthesis make up the genome replication programme of *L. major*, with a single predominant S-phase region of initiation in each chromosome supplemented by subtelomeric DNA synthesis activity that appears to extend beyond S phase.

## Results

### Detection of subtelomeric DNA replication throughout the cell cycle of *L. major*

Previous MFA-seq analysis, which detected a single region of replication initiation per chromosome in *L. major* and *L. mexicana* (*Marques et al., 2015*), is potentially limiting because it relies on cell sorting to enrich for S phase (replicating) and G2 or G1 (presumably non-replicating) cells, then inferring regions of DNA synthesis by comparing sequence read depth in the former relative to the latter. As a result, DNA replication very late in S phase, or that extends beyond the conventional S phase, would escape detection. To test this, we modified the MFA-seq approach in order that DNA content enrichment could be calculated in replicating cells relative to naturally occurring non-replicating cells. Mapping origins of replication by calculating DNA enrichment in exponentially growing cells versus cells in a stationary state has been used successfully in bacteria (*Ivanova et al., 2015*) and yeast (*Müller et al., 2014*). Thus, we reasoned that *L. major* in stationary phase could also serve as a non-replicative control for normalisation in MFA-seq analysis compared with cells that are growing exponentially. To test this, we used flow cytometry to compare DNA content of exponentially growing and stationary phase cells (*Figure 1A*) and, in addition, compared the capacity of the two cell populations to incorporate thymidine analogues (*Figure 1B,C*). Flow cytometry showed that the proportion of cells in S phase (with DNA content between 1C and 2C) was substantially lower in a population of stationary cells compared with exponentially growing cells (*Figure 1A*). Concomitantly, stationary phase cells, unlike exponentially growing cells, incorporated very low levels of IdU or EdU (*Figure 1B,C*). Thus, *L. major* cells in stationary phase perform little detectable DNA synthesis and were therefore deemed suitable to be used as a non-replicative sample in MFA-seq analysis.

Next, we compared MFA-seq profiles using read depth ratios from FACS-sorted early S (ES) and G2 cells, and from exponentially growing cells (EXP) and stationary (STA) cells (*Figure 1D*; *Figure 1—figure supplement 1* details the sequence data used and its processing for subsequent analyses). As reported previously (*Marques et al., 2015*), mapping the ratio of ES/G2 sequence reads revealed a single peak in each chromosome, with all peaks localising to the boundaries of the multigene transcription units. The same peaks were also apparent in EXP/STA read mapping but, strikingly, further regions of read enrichment were now detected that localised towards the chromosome ends (*Figure 1D*). Subtelomeric enrichment in EXP/STA could not be explained by loss of read coverage in these regions of the chromosomes in STA cells (*Figure 1—figure supplement 2*), and though the MFA-seq signal was more clearly detected when the data were analysed by Z score (*Figure 1—figure supplements 3*, *4* and *5*), the same pattern was discernible when analysing copy-number variation (*Figure 1—figure supplements 3* and *4*). To explore these new regions of predicted DNA replication initiation further, we compared read depth in exponentially growing cells sorted into G1, ES, late S (LS) and G2/M phases of the cell cycle (see gating strategy in *Figure 1A*) relative to STA cells. The pronounced ES/G2 MFA-seq peaks were only clearly seen in the ES/STA MFA-seq mapping (*Figure 1D*, *Figure 1—figure supplement 5*), whereas subtelomeric signal was seen in several chromosomes from the cell cycle-sorted DNA relative to STA. In ES/STA MFA-seq mapping, the amplitude of subtelomeric MFA-seq signal was lower than the more central MFA-seq peak in each chromosome and, indeed, was most clearly seen in the larger chromosomes. However, the subtelomeric MFA-seq signal was notably stronger in G1/STA and G2/STA MFA-seq mapping of all chromosomes, where the central MFA-seq peak was much less pronounced. Taken together, these data indicate two things. First, sites of subtelomeric DNA synthesis, predicted by change in sequence copy number, appear spatially separate from previously mapped, predominantly chromosome-central regions of DNA replication initiation (*Marques et al., 2015*) and can be found in virtually every

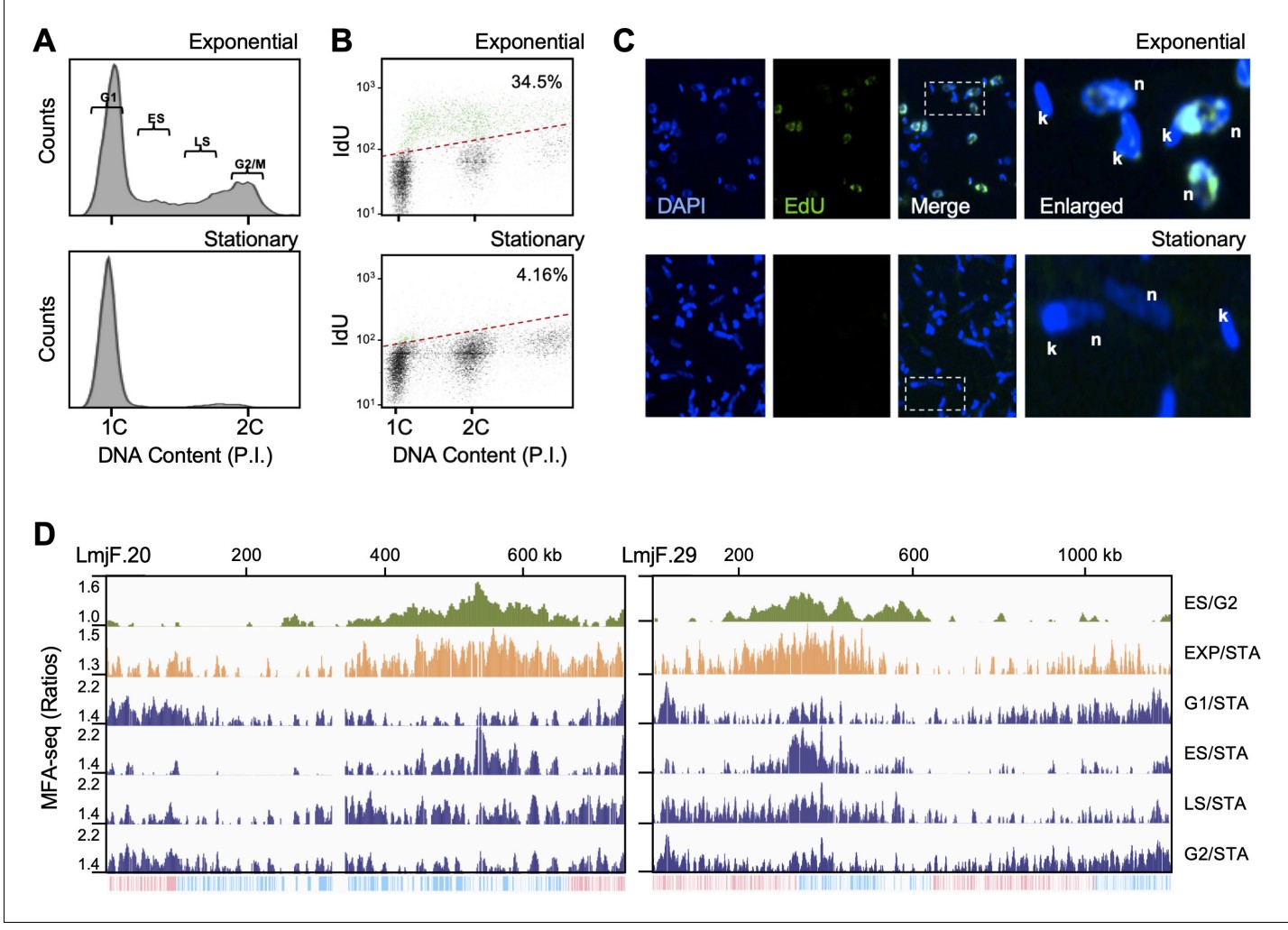

**Figure 1.** Detection of DNA replication initiation throughout the cell cycle in *L. major*. (A) DNA content analysis by FACS of cells in exponential growth or in stationary phase; DNA was stained with Propidium Iodide (PI). 1C and 2C indicate single and double DNA content, respectively; gates used to sort exponentially growing cells into G1, early S (ES), late S (LS) and G2/M enriched populations are indicated. (B) Representative density plots from flow cytometry analysis to detect DNA synthesis in exponentially growing and stationary phase cells; cells were incubated with IdU for 30 min and IdU fluorescence was detected under denaturing conditions; 30,000 cells were analysed; 1C and 2C indicate single and double DNA content, respectively; dashed red lines indicate the threshold used to discriminate negative (black dots) from IdU-positive (blue dots) events; inset numbers indicate total percentage of IdU-positive events relative to the whole population. (C) Cells in exponential growth or in stationary phase were pulsed with 10 μM EdU for 1 hr, subjected to click reaction and then visualised by confocal microscopy and DAPI staining; n and k indicate DNA from the nucleus and kinetoplast, respectively. (D) MFA-seq profile for the indicated chromosomes; from top to bottom: first track shows previously published MFA-seq profile in ES using G2/M cells for normalisation *Marques et al., 2015*; second track shows MFA-seq profile of exponentially growing cells, not sorted, using stationary (STA) cells for normalisation; the remaining tracks show MFA-seq profile for exponentially growing cells, sorted in the indicated cell cycle stages, using STA cells for normalisation; MFA-seq signal is shown as ratios of reads for all samples (y-axes; enriched populations indicated to the right), plotted in five kbp windows across the length of the chromosomes (x-axes; chromosome sizes are indicated); the bottom track in each panel indicates annotated CDSs (blue: transcribed from left to right; pink: transcribed from right to left).

The online version of this article includes the following figure supplement(s) for figure 1:

**Figure supplement 1.** Distribution of reads per bin.

**Figure supplement 2.** Stationary phase sequencing coverage.

**Figure supplement 3.** Overview of data analysis.

**Figure supplement 4.** Comparing MFA-seq expressed as ratio and Z scores in the context of aneuploidy.

**Figure supplement 5.** MFA-seq profile for all chromosomes of *Leishmania major*.

*L. major* chromosome. Second, whereas a single replication initiation region in each *L. major* chromosome is activated early in S phase, predicted subtelomeric DNA replication initiation can be detected in all enriched stages of the *L. major* cell cycle.

## MFA-seq reveals the programme of replication timing in the *L. major* genome

Analysis of the EXP/STA MFA-seq profile on individual chromosomes suggested the possibility of a programme of DNA replication initiation throughout the cell cycle (*Figure 1D*, *Figure 1—figure supplement 5*). In ES and LS, MFA-seq signal appeared to be found mainly in the internal regions of the chromosomes (completely overlapping previously mapped ES/G2 MFA-seq peaks), while signal was strongest proximal to the chromosome ends during G1 and G2/M. To test this possibility further, we first performed meta-analyses, comparing MFA-seq profiles across all chromosomes in EXP/STA cells (*Figure 2A*) and across different stages of the cell cycle (*Figure 2B*). In both EXP and ES cells, MFA-seq was seen as peaks of very consistent amplitude and width that fell into three clusters, one where the peak was found centrally in the chromosomes, and two others where the peak was displaced towards the right or left ends of the chromosomes. There was no obvious separation of peak localisation into the three clusters based on chromosome size (*Figure 2C*). In LS cells (*Figure 2B*), the MFA-seq peaks in each cluster had increased in width, again to a very consistent extent, with half of the peaks from the less central clusters having merged with the chromosome ends. These data are consistent with bidirectional progression of replication forks emanating from a single predominant region in each chromosome and suggest considerable consistency in rate or pattern of fork movement irrespective of initiation site. Finally, in G2/M and G1 cells, the three clusters overlapped to a much greater extent, with MFA-seq signal again showing a consistent pattern across the genome, but here the signal peaked at the chromosome ends and reduced in amplitude towards the interior of the molecules (*Figure 2B*). These data may be explained by DNA replication in *L. major* following a programme in which synthesis of a new chromosome initiates at a single predominant region in the interior of each chromosome in early S phase and progresses at a consistent rate towards the chromosome ends, but with DNA replication continuing even as cells navigate from LS through G2/M until they enter G1. Alternatively, the profiles may suggest a bimodal DNA replication programme, with all chromosomes initiating DNA replication from internal regions in ES phase as well as using DNA replication initiation that derives from the telomeres or from telomere-proximal regions, either late in S phase or in G2–G1 phases.

Since a single major MFA-seq peak was seen in each chromosome in ES cells, the majority of DNA replication is predicted to initiate from a single region in each case. Given the range of chromosome sizes in the *L. major* genome (~0.25–2.4 Mb), a prediction of the mapping is that the time to complete chromosome replication will be size-dependent. To test this prediction, we estimated the extent of each chromosome that is replicated in early S phase by comparing the ES/STA MFA-seq ratios with G2/STA ratios, revealing a greater extent of DNA replication in smaller chromosomes relative to the larger (*Figure 2D*). To explore this further, we separated the chromosomes into three size groupings and significant differences in replication timing were clearly seen (*Figure 2E*).

## Chromatin and base composition differ at sites of core and subtelomere chromosome replication initiation

To date, our understanding of what DNA sequence or chromatin elements might dictate Leishmania DNA replication initiation is limited, and so we asked how the DNA replication pattern across the *L. major* cell cycle correlates with known chromatin and DNA features (*Figure 3*). Specifically, we examined the distribution of MFA-seq signal relative to sites of enrichment for acetylated histone H3 (AcH3; associated with transcription initiation) (*Thomas et al., 2009*), ß-D-glucosyl-hydroxymethyluracil (base J; associated with transcription termination) (*van Luenen et al., 2012*; *Reynolds et al., 2014*) and kinetoplastid kinetochore protein 1 (KKT1, which is most strongly enriched at a single site in each chromosome of Leishmania) (*Garcia-Silva et al., 2017*). This analysis revealed a difference between the chromosome-internal and subtelomere DNA replication reactions. As exemplified by chromosome 35 (*Figure 3A*), simultaneous co-localisation of AcH3, base J and KKT1 was found at the single MFA-seq peak seen in ES cells in every chromosome. Surprisingly, this co-localisation included MFA-seq peaks at six sites where the surrounding multigene transcription units converge,

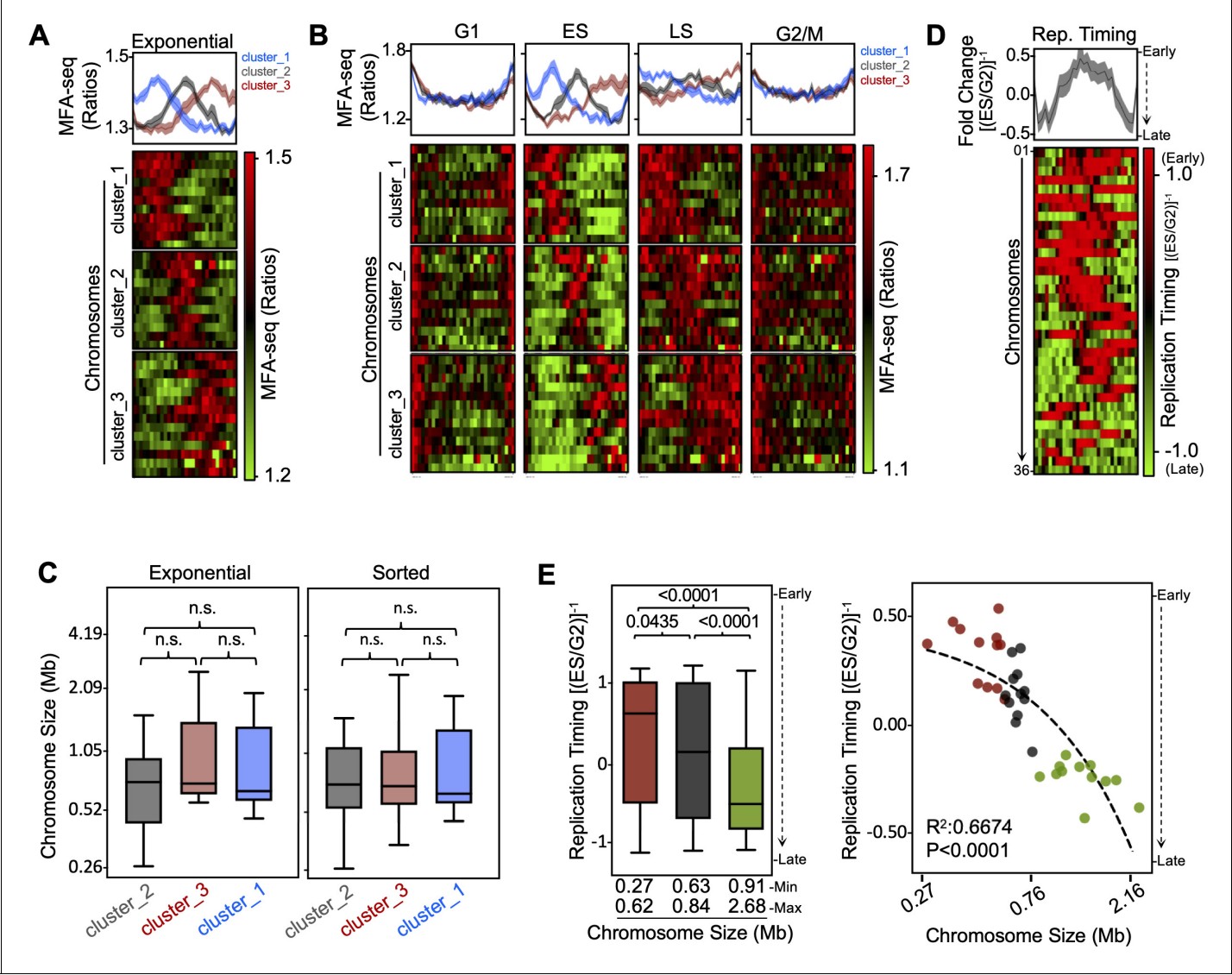

**Figure 2.** Changes in MFA-seq signal profile throughout the cell cycle. MFA-seq profile in (A) exponentially growing or (B) sorted cells in the indicated phases of cell cycle, in both cases relative to stationary phase (STA) cells; panels at the top represent global MFA-seq signal for the whole genome; colourmaps indicate MFA-seq profile for individual chromosomes (each row). Chromosomes were grouped by k-means clustering, using deepTools. Chromosomes in each cluster from exponential cells: *cluster_1*–01, 03, 05, 08, 19, 20, 22, 23, 27, 28, 32; *cluster_2*–02, 09, 10, 12, 15, 16, 17, 18, 24, 25, 26, 31, 33, 36; *cluster_3*–04, 06, 07, 11, 13, 14, 21, 29, 30, 34, 35; chromosomes in each cluster from sorted cells: *cluster_1*–01, 02, 03, 05, 08, 18, 19, 20, 22, 23, 25, 27,28,32; *cluster_2*–09, 10, 12, 15, 16, 17, 24, 26, 31, 33, 36; *cluster_3*–04, 06, 07, 11, 13, 14, 29, 21, 30, 34, 35. (C) Replication timing was estimated by calculating the fold change between MFA-seq signal (ratio) from ES cells and MFA-seq signal (ratio) from G2 cells relative to STA; panels at the top represent global MFA-seq signal for the whole genome; colourmaps indicate MFA-seq profile for individual chromosomes (each row). (D) Chromosome size is plotted for each cluster shown in A and B; differences were analysed with a Kruskal-Wallis test; n.s., not significant. (E) Mean replication timing (as calculated in C) from each chromosome is plotted; left, chromosomes were grouped according to their sizes (small, medium and large) and *P* values, analysed with a Kruskal–Wallis test, are shown for each indicated pair; right, mean values were plotted for each individual chromosome, with R[2] and *P* values for linear regression shown.

and at the telomere of chromosome 1; in each case these are locations of transcription termination, where AcH3 enrichment is not predicted (***Thomas et al., 2009***), but modest AcH3 enrichment was seen in all cases (***Figure 3B***, ***Figure 1—figure supplement 5***). Meta-analysis of all chromosomes, testing correlation of MFA-seq signal in different cell cycle stages around loci where the AcH3, base J and KKT1 ChIP datasets overlap, revealed peaks of considerable consistency in amplitude and width in EXP and ES cells, which diminished as cells progressed to LS (***Figure 3C***). These data

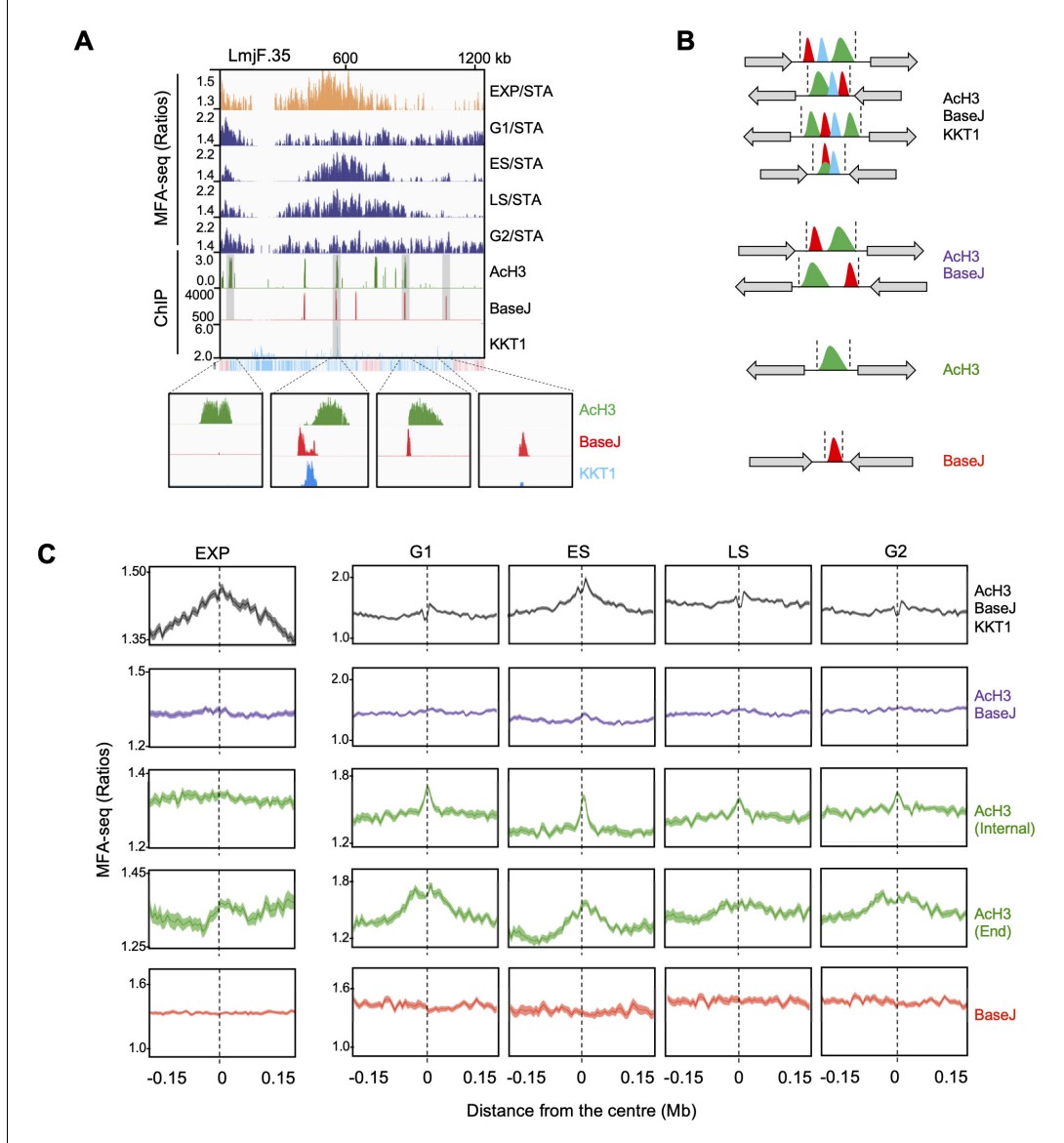

**Figure 3.** Changes in MFA-seq signal profile throughout the cell cycle are associated with distinct chromatin features. (A) A representative region of chromosome 35, showing MFA-seq signals in the indicated phases of cell cycle, as compared to the positioning of AcH3, BaseJ and KKT1 enriched sites; the bottom track indicates annotated CDSs (blue: transcribed from left to right; pink: transcribed from right to left); for improved visualisation, segments of interest are shaded and magnified at the bottom. (B) Schematic illustration of regions of transcription initiation or termination, containing the indicated combination of chromatin markers; gray arrows indicate all possible configurations of surrounding multigene transcription direction; coordinates defining the boundaries of these regions were taken from *Lombraña et al., 2016* and manually curated. (C) Metaplots of global MFA-seq signal, in exponential cells or at the indicated phases of the cell cycle, relative to STA cells ± 0.15 Mb from the centre of regions containing the indicated combination of chromatin marks; light-coloured areas around the lines indicate the standard error of the mean (SEM).

The online version of this article includes the following figure supplement(s) for figure 3:

**Figure supplement 1.** Changes in MFA-seq signal profile throughout the cell.

**Figure supplement 2.** MFA-seq signal near chromosome's ends.

**Figure supplement 3.** MFA-seq of exponentially growing cells.

**Figure supplement 4.** DNA sequence within the regions containing the indicated combination of the chromatin markers was subject to MEME analysis; top three motifs (most abundant and lowest *p-value*) found within each region are shown.

suggest that the simultaneous presence of these three genome factors is a local driver for coordinated DNA replication initiation at a single region in each chromosome in ES but does not promote DNA replication initiation in other cell cycle stages.

Meta-analysis of MFA-seq at all sites where base J was found, alone or in combination with AcH3, showed that the modified base itself has little correlation with DNA replication in any cell cycle stage (*Figure 3C*; *Figure 3—figure supplement 1*). In contrast, MFA-seq signal was seen around sites of AcH3 enrichment in all cell cycle stages analysed (*Figure 3C*), suggesting that DNA replication in this chromatin context is ubiquitous throughout the cell cycle. Because chromatin environment and DNA sequence near telomeres might differ from the internal regions of chromosomes, we separately analysed MFA-seq signal around AcH3 sites near chromosome ends (<10 kb from telomeres) and those located elsewhere in the genome (>10 kb from telomeres). In both cases, MFA-seq signal was seen around sites of AcH3 enrichment in all phases of the cell cycle (*Figure 3C*; *Figure 3—figure supplement 1*). However, MFA-seq peaks around non-telomeric AcH3 regions were notably smaller in amplitude and narrower in width compared with the pronounced MFA-seq signal around telomere-proximal AcH3 sites. These data provide confirmation of a genome-wide association between AcH3 sites and DNA replication initiation throughout the cell cycle, but it appears that while AcH3 sites are associated with DNA replication initiation in the chromosome subtelomeres, fewer AcH3 sites are used in the chromosome interiors. Therefore, the genomic context where this histone modification is found correlates with distinct local replication initiation activity, perhaps because of unrecognised features at these sites in the subtelomeres. Alternatively, stronger subtelomeric MFA-seq signal may indicate DNA replication initiation emanating from the telomeres, and not necessarily from AcH3 sites.

To further explore the extent of DNA replication initiation near end-proximal AcH3 sites, we first mapped sequence ratios around the subtelomeres in ES, LS and G2-enriched cells relative to G1-enriched, rather than STA cells (*Figure 3—figure supplement 2A*). Conversely, sequence ratios in G1 were also calculated relative to ES, LS and G2 enriched populations (*Figure 3—figure supplement 2B*). In both cases, increased copy-number variation in the proximity of chromosome ends was seen in G1 relative to ES, LS and G2. This further demonstrates that these regions serve as DNA replication initiation sites, and that such sites seem to have a relatively higher usage during G1. Next, we compared MFA-seq signal in EXP/STA cells using both copy-number change and Z-score analyses. Irrespective of the analysis strategy, telomere-proximal signal, although less pronounced than the central enriched regions, was apparent in several chromosomes, though not in the majority (*Figure 3—figure supplement 3*). Consistent with this, meta-analysis of MFA-seq signal in EXP cells showed only minor signal enrichment around telomeric AcH3 sites (*Figure 3C*). Therefore, these data indicate that subtelomere DNA replication initiation is mainly not activated during ES but is a persistent activity throughout the enriched cell cycle populations. Altogether, these observations indicate the importance of local chromatin environment in modulating *L. major* DNA replication initiation in a temporal manner.

## Distinct strand asymmetry and G4 distribution at sites of core and subtelomeric DNA replication initiation

To ask if genetic elements might be associated with DNA replication initiation in *L. major*, we next sought to analyse the patterns of sequence composition within and around the loci containing the above combinations of AcH3, base J and KKT1 enrichment. First, we tested for the existence of specific sequence motifs associated with the distinct patterns of predicted DNA replication initiation around these sites. As expected, MEME analysis retrieved motifs that have been associated with transcription initiation and termination, known to take place at these sites (G/T rich motifs and polypyrimidine tracks; *Figure 3—figure supplement 4*; *Clayton, 2019*). We could not find any sequence signature that correlated specifically with regions of DNA replication initiation, or that differed between the regions used during ES phase or throughout the cell cycle.

Next, we examined the patterns of intra-strand base composition skews (excess of G over C = [(G-C)/(G+C)]; excess of T over A = [(T-A)/(T+A)]). In many organisms, G skew and, to a lesser extent, T skew values switch polarity around replication initiation sites, with the leading strand usually associated with positive skews values (*Xia, 2018*; *Huvet et al., 2007*; *Baker et al., 2012*). With this in mind, we might expect to detect a predominance of negative G and T skew values upstream from the main DNA replication initiation regions mapped in ES cells by MFA-seq and co-localising with

AcH3/base J/KKT1 sites. Conversely, positive G and T skew values would predominate downstream from these sites. Our analysis did not indicate such a broad replication-associated pattern, but instead revealed that the major correlation of base skew in *L. major* is with transcription direction, in which the coding strand is mainly associated with positive G skew and negative T skew values (*Figure 4A*), as previously reported (*El-Sayed et al., 2005*; *McDonagh et al., 2000*). However, a closer examination of the skew patterns in the central area of the ES MFA-seq peaks revealed a highly localised pattern of polarity change, resembling the wider pattern seen around origins of replication in other organisms (see zoomed region in chromosomes 30 and 31; *Figure 4B*). This local

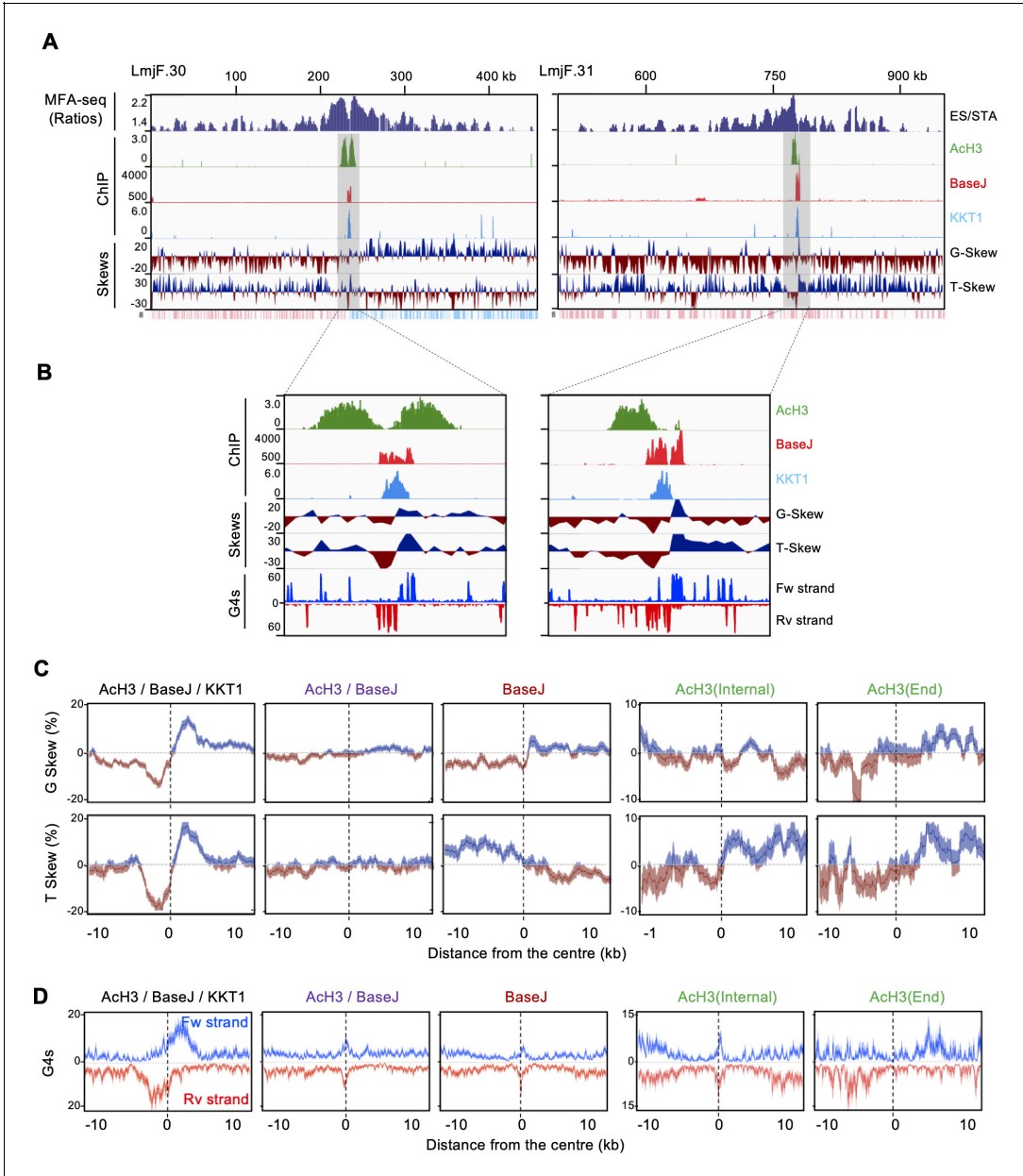

**Figure 4.** Predicted DNA replication initiation regions are related to distinct patterns of G skew, T skew and G4 accumulation. (**A**) Representative regions of chromosomes 30 and 31, showing MFA-seq signals in ES/STA, compared to the positioning of the indicated chromatin markers; G and T skews are also shown; the bottom track in each panel indicates annotated CDSs (blue: transcribed from left to right; pink: transcribed from right to left). (**B**) Shaded areas in A are magnified, and now include G4 peaks distribution for both forward (Fw strand) and reverse (Rv strand) DNA strands. (**C** and **D**) Metaplots of global G and T skews (**C**) and G4 distribution (**D**) ± 10 kb from the centre of regions containing the indicated combination of chromatin markers; light-coloured areas around lines indicates SEM.

change, from negative to positive in the G and T skew values across the centre of the regions of AcH3/base J/KKT1 enrichment, could be seen even when two adjacent transcription units are transcribed in the same direction (see zoomed region in chromosome 31, which is entirely transcribed unidirectionally; *Figure 4B*). Accordingly, meta-analysis revealed the same localised pattern of polarity change, for both G and T skew, when every ES MFA-seq site was compared (*Figure 4C*). A similar pattern was seen across AcH3 sites near chromosome ends, although both the amplitude and sharpness of the polarity change in base skew was less pronounced than that seen around AcH3/base J/ KKT1 sites. In contrast, only T skew showed the same global polarity change around chromosome-internal AcH3 sites (*Figure 4B*). Finally, we did not find any obviously related profile of base skew polarity change around either AcH3/base J or base J sites (*Figure 4C*). These data provide clear evidence for base composition bias, consistent with DNA replication initiation, at a central region in each of the 36 *L. major* chromosomes. In addition, differences in base composition around core and subtelomeric AcH3 sites are indicative of extensive DNA replication initiation at the latter but not the former.

Next, we sought to examine the potential correlation between G-quadruplexes (G4s) and the distinct DNA replication initiation activities we see across the *L. major* genome. G4s are secondary DNA structures that can arise in single-stranded, guanine-rich DNA and have been implicated in various biological processes in eukaryotic cells, including DNA replication (*Cayrou et al., 2015*; *Besnard et al., 2012*). G4s have been predicted to colocalise with SNS-seq signal in *Leishmania* (*Lombraña et al., 2016*), but their relationship with DNA replication initiation regions mapped by MFA-seq has not been explored. Thus, we examined the distribution of experimentally mapped G4s (*Marsico et al., 2019*) around all sites with the differing combinations of AcH3, base J and KKT1 binding (*Figure 4D*). Our analyses showed a strong positional preference for G4 distribution around the midpoint where G and T skews change polarity across the AcH3/base J/KKT1 sites (see zoomed regions in chromosomes 30 and 31, *Figure 4B*). Upstream of the midpoint of these sites, a grouping of G4 peaks could be seen on the reverse DNA strand, while downstream of the midpoint a similar grouping was seen on the forward strand. Meta-analysis showed that these localised patterns of G4 distribution were reflected in every ES MFA-seq mapped peak, with a striking correlation relative to the base skew polarity change (*Figure 4D*). A similar global pattern of G4 distribution, although more diffuse, was observed around subtelomeric AcH3 sites (*Figure 4D*). At all these predicted sites of DNA replication the G4 distribution differed from the pattern seen at chromosome-internal AcH3 sites, as well as at AcH3/base J and base J sites, where G4s were arranged as sharp peaks on both the forward and reverse DNA strands that aligned at the central point of enrichment (*Figure 4D*). Altogether, these observations indicate that the region of *L. major* DNA replication initiation predicted by MFA-seq have imprinted, or have been shaped by, distinct local profiles of DNA intrastrand asymmetries and arrangement of G4s. Moreover, the patterns of these genome features reflect the predicted differences, in space and time, of DNA replication activity at chromosome-internal and subtelomeric regions.

Multigene transcription surrounding AcH3/base J/KKT1 sites can have three distinct configurations: the transcription units diverge (transcription initiation of both units within the locus), converge (transcription termination) or are arranged head to tail (combined transcription initiation and termination) (*Figure 5A*). To ask about the relationship between replication initiation and transcription activity, we plotted the G/T skews and G4 profile around each of these configurations. As expected, at all sites a pronounced MFA-seq peak was seen when comparing ES or EXP cells with STA cells, with no evidence for differential DNA replication initiation activity between them (*Figure 5B*). However, by separating these sites, it became clear that the patterns of G skew, T skew (*Figure 5C*) and G4 distribution (*Figure 5D*) were mainly dictated by the leading strand of the predicted bidirectional DNA replication fork, not by the lagging strand, irrespective of the surrounding transcription direction. Moreover, the analysis suggested that leading strand DNA replication and transcription are always co-directional on the same template DNA strand around these chromosome-central putative origins. Therefore, this analysis advances the correlation between DNA replication initiation activity and local DNA strand asymmetry and G4 distribution and suggests the existence of selective pressure to avoid clashes between DNA replisome and RNA polymerase activities at regions of Leishmania DNA replication initiation.

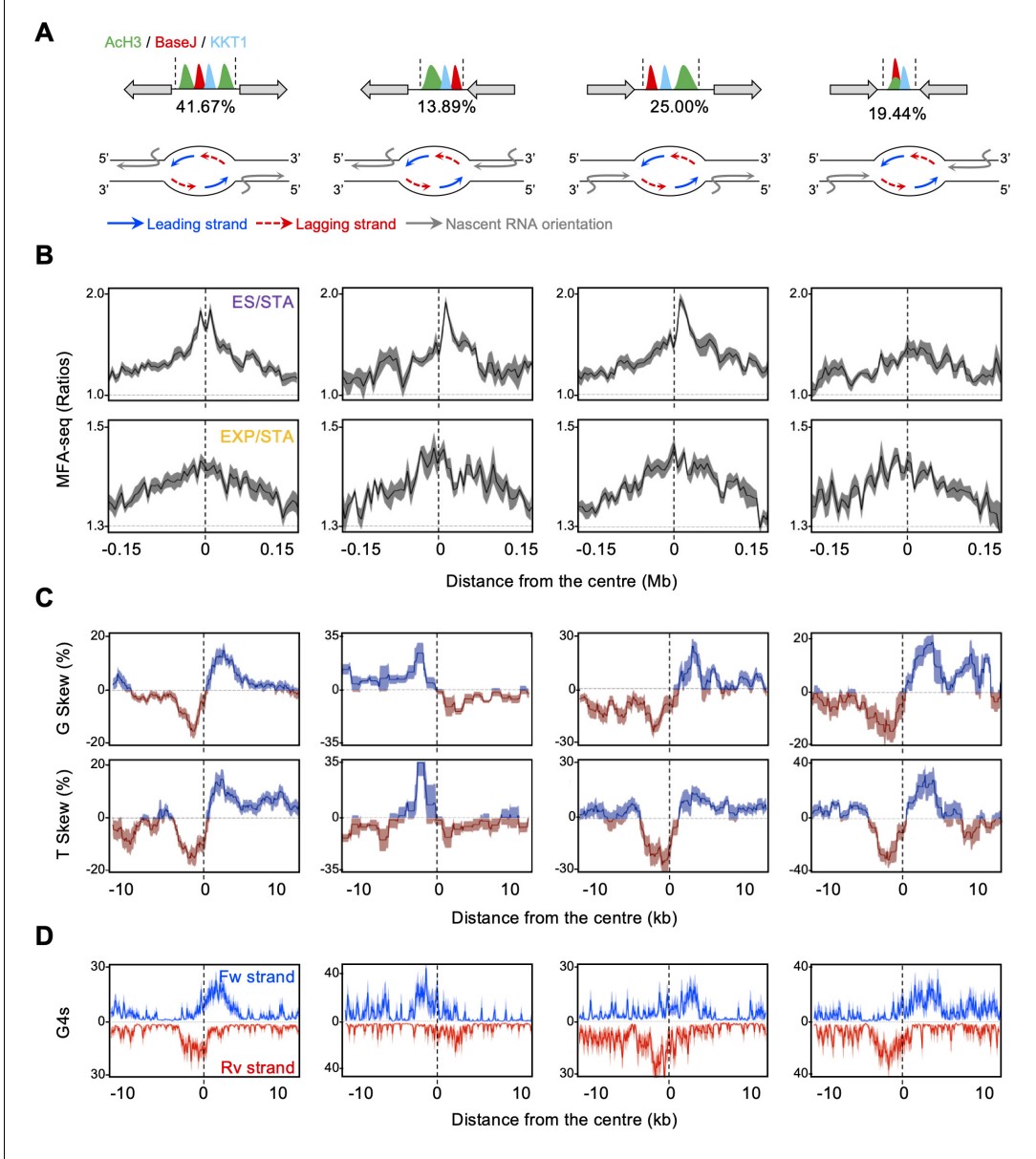

**Figure 5.** Local patterns of G and T skews at early S phase replication initiation are associated with co-directional transcription and DNA replication activity. (**A**) Top: a schematic illustration of sites containing simultaneous deposition of AcH3, BaseJ and KKT1, with gray arrows indicating all possible configurations of surrounding multigenic transcription direction; numbers below each scheme indicates the proportion of each configuration. Bottom: schematic illustration of transcription and replication organisation on the DNA strands at each of these sites. (**B**), (**C** and **D**) MFA-seq profile (ES/STA and EXP/STA), base skews (G and T) and G4 pattern (forward and reverse DNA strands) are shown, respectively, for each of these sites as metaplots.

## DNA replication initiation around chromosome ends is more susceptible to replication stress

The data described above suggest differences in the timing and location of chromosome-internal and subtelomeric DNA replication initiation, but do not address if they use shared or separate machineries. To begin to answer this question, we asked if and how replication stress could affect the DNA replication programme in *L. major*. To do so, we performed MFA-seq analysis on cells sorted into ES and G2/M after release from replication arrest with 5 mM hydroxyurea (HU), which depletes the intracellular dNTP pool (*Figure 6A*). To examine DNA replication, we performed MFA-seq, comparing read depth relative to STA cells. We first searched for the appearance of new replication initiation sites, perhaps indicative of activation of dormant origins. At only three sites could

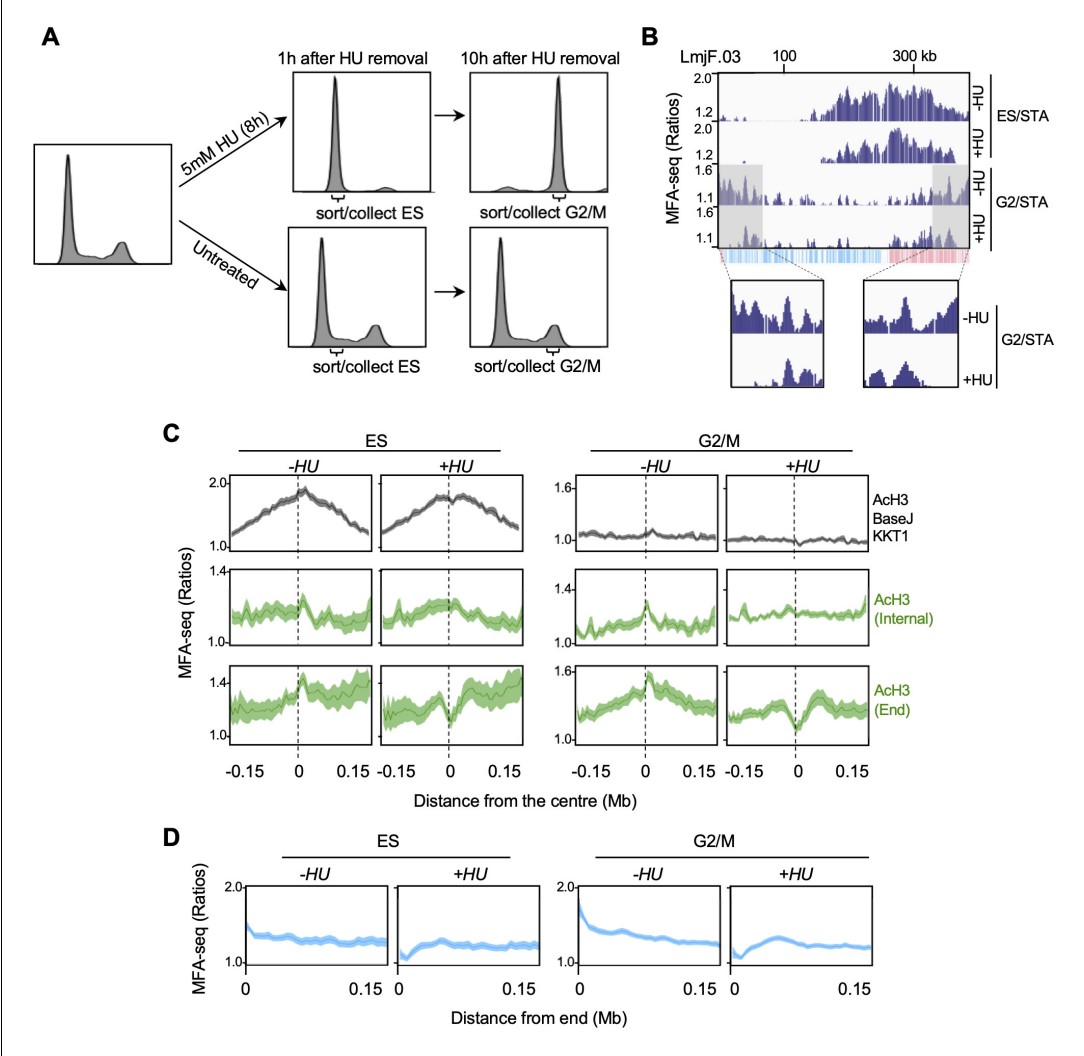

**Figure 6.** DNA replication close to chromosome ends is more sensitive to replication stress than replication at chromosome-central loci. (A) Cells were left untreated or treated with hydroxyurea (HU) and, at the indicated times after HU removal, sorted and the indicated cell cycle stages were collected; plots show FACS profiles of the cell populations after PI staining. (B) MFA-seq profile for the indicated phases of the cell cycle relative to STA, with or without prior HU treatment, in the indicated region of chromosome 3; shaded areas are magnified in the boxes at the bottom to highlight the differences in MFA-seq signal at regions close to chromosome end, upon HU treatment; the bottom track indicates annotated CDSs (blue: transcribed from left to right; pink: transcribed from right to left). (C) Metaplots of global MFA-seq signal, in the indicated phases of the cell cycle,±0.15 Mb from the centre of regions containing the indicated combination of chromatin markers; lines represent mean read ratios and light-coloured areas indicate SEM. (D) Metaplots of global MFA-seq signal for the indicated phases of cell cycle relative to STA, with or without prior HU treatment, across 0.15 Mb of all chromosome ends.

The online version of this article includes the following figure supplement(s) for figure 6:

**Figure supplement 1.** MFA-seq profile of cells at the indicated phases of cell cycle, with or without prior HU treatment; red arrows indicate sites with increased MFA-seq signal in HU treated cells, as compared with not treated cells.

**Figure supplement 2.** Extended data for *Figure 6*.

we detect an extremely mild increase in MFA-seq signal in ES or G2/M cells after HU treatment compared with non-HU treated cells (red arrows in *Figure 6—figure supplement 1*). Though all these sites were located at the boundaries of multigene transcription units, like the single predominant ES MFA-seq peak in each chromosome, these data suggest that *L. major* has a very limited capacity to initiate DNA replication initiation from other loci in ES phase, though when it occurs such a reaction also predominantly localises to sites of transcription initiation or termination.

Next, we looked for changes in the previously detected sites of DNA synthesis across the genome after HU treatment. First, we examined DNA replication in ES cells around the single AcH3/base J/KKT1 site in each chromosome. Mapping to each chromosome (*Figure 6B*), as well as meta-analysis of MFA-seq signal in all chromosomes (*Figure 6C*), did not reveal any change in MFA-seq peak profile in the HU treated cells, suggesting that DNA replication emanating from these regions is largely unaffected by this level of HU. In contrast, we saw pronounced changes in subtelomere replication, since a decrease in MFA-seq signal at the chromosome ends during G2/M after release from HU was detected in individual chromosomes (*Figure 6B*). Meta-analysis revealed the extent of this perturbation. First, a marked loss of MFA-seq signal, visible as early as in ES, was seen around the subtelomeric AcH3 sites upon HU treatment (*Figure 6C*; *Figure 6—figure supplement 2*), whereas no such clear effect was seen around chromosome-internal AcH3 sites (*Figure 6C*). Second, meta-analysis of MFA-seq signal at the ends of every chromosome (*Figure 6D*) showed that the HU-induced loss of the replication signal extended for ~50 kb. Unsurprisingly, given the lack of clearly detectable MFA-seq predicted DNA replication at AcH3/base J or base J sites, meta-analysis did not show any significant changes in global MFA-seq signal at either of these type of loci (*Figure 6—figure supplement 2*). These observations reveal that not all DNA replication initiation sites are equally affected by replication stress, with replication activity proximal to chromosome ends being more susceptible to HU treatment than replication from chromosome-internal regions. Moreover, the data reinforce the suggestion of differences in the nature of DNA replication around subtelomeric AcH3 sites and chromosome-internal AcH3/base J/KKT1 sites.

## Replication initiation at chromosome ends requires RAD9 and HUS1

To begin to explore the factors needed for Leishmania genome duplication, we wondered if an explanation for the newly described subtelomeric DNA replication might be that, in this chromosome environment, DNA replication is under constitutively higher levels of stress when compared with DNA replication from the chromosome-internal AcH3/base J/KKT1 regions. If so, we reasoned that subtelomeric DNA replication might have a greater dependence on the replication stress response machinery. RAD9 and HUS1 are two components of the 9-1-1 clamp, which is a cell cycle checkpoint complex required for replicative DNA synthesis in *L. major* (*Damasceno et al., 2016*; *Damasceno et al., 2018*). As a result, we asked if RAD9 or HUS1 deficiency would alter the profile of *L. major* DNA replication. To do this, we isolated ES- and G2/M-phase cells from HUS1 and RAD9 heterozygous (+/-) mutants and performed MFA-seq as before. Strikingly, both visual inspection of individual chromosomes (*Figure 7A*) and meta-analysis (*Figure 7B and C*, *Figure 7—figure supplement 1*) showed a dramatic decrease in MFA-seq signal around subtelomeric AcH3 sites and proximal to the chromosome ends in RAD9$^{+/-}$ cells, in both ES and G2/M. In contrast, no obvious change in MFA-seq signal was seen around chromosome-internal AcH3 sites in G2/M RAD9$^{+/-}$ cells and, even more strikingly, no change in MFA-seq signal around AcH3/base J/KKT1 was detected in the RAD9$^{+/-}$ cells (*Figure 7B*). In HUS1$^{+/-}$ cells, a milder decrease in MFA-seq signal was observed, which was apparent only for the subtelomeric AcH3 sites during G2/M (*Figure 7B* and *Figure 7C*). Consistently, decreased MFA-seq signal near telomeres was also seen when directly comparing sequence abundance in G2-enriched populations from both mutants and wild type (WT) cells (*Figure 7—figure supplement 2*). No alteration in the MFA-seq signal was detected around AcH3/base J or base J sites in either of the mutants (*Figure 7—figure supplement 1*). These data indicate DNA replication initiation close to chromosome ends, but not at the distinct chromosome-internal initiation regions, requires the action of the 9-1-1 complex.

## Discussion

In this study, we have analysed the orchestration of Leishmania genome replication in space and time. Our data show that genome duplication predominantly starts from a single region in every chromosome in early S phase, proceeding towards the chromosome ends, consistent with the observed size dependence of *L. major* chromosome replication timing. However, replication of the chromosomes is rarely completed by the end of S phase, and instead either continues from the same sites as cells transit through G2/M back to G1, or new sites of DNA replication initiation are activated in the chromosome subtelomeres during late S phase or in G2/M and G1 stages of the cell

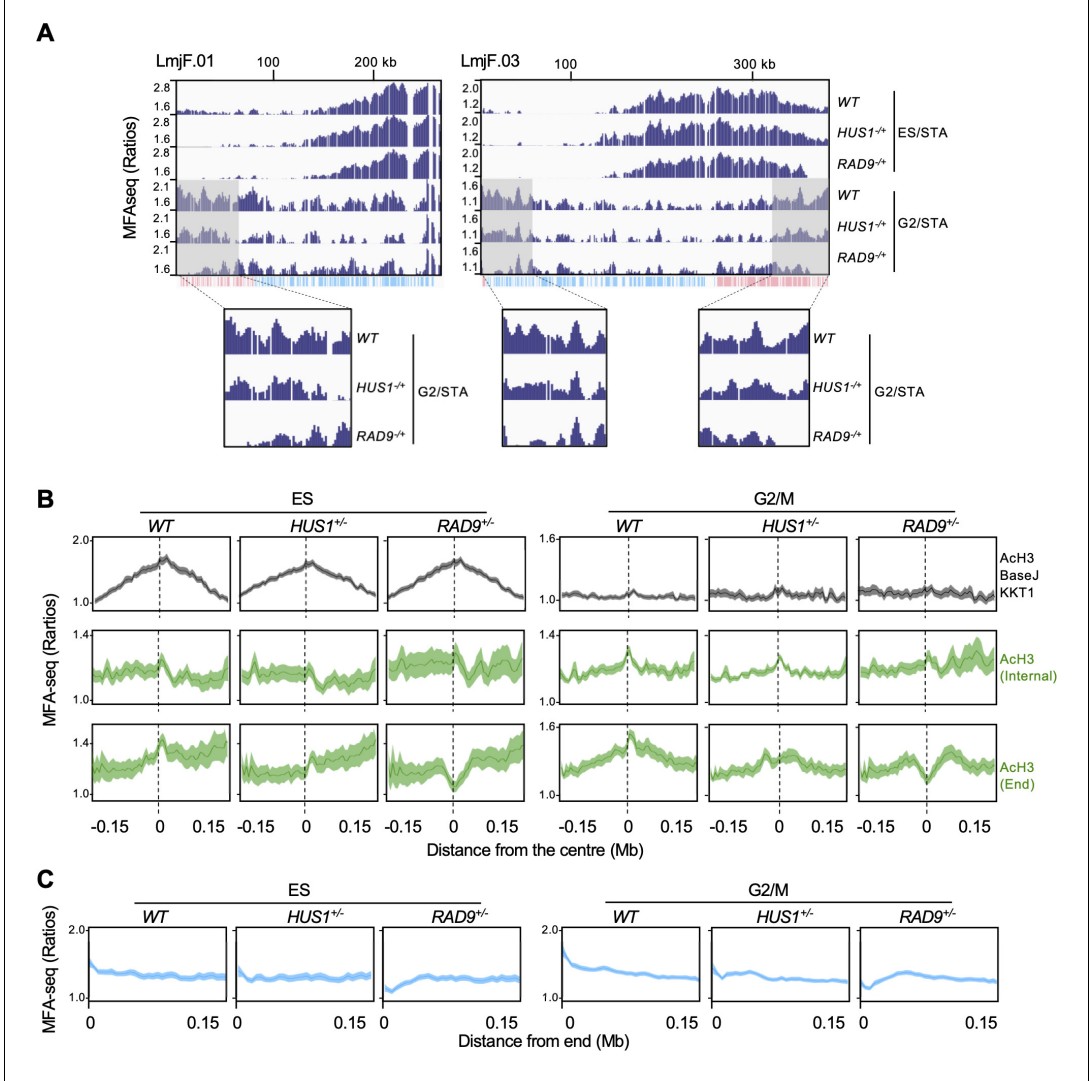

**Figure 7.** RAD9 and HUS1 are required for DNA replication near chromosome ends. (**A**) MFA-seq profile of wild type (*WT*), *RAD9*⁺ᐟ⁻ and *HUS1*⁺ᐟ⁻ cells at the indicated phases of cell cycle across chromosomes 1 and 3; shaded areas are magnified in the boxes at the bottom to highlight the differences in MFA-seq signal at regions close to chromosome ends; the bottom track indicates annotated CDSs (blue: transcribed from left to right; pink: transcribed from right to left). (**B**) Metaplots of global MFA-seq signal of *WT*, *RAD9*⁺ᐟ⁻ and *HUS1*⁺ᐟ⁻ cells, in the indicated phases of the cell cycle,±0.15 Mb from the centre of regions containing the indicated combination of chromatin markers. (**C**) Metaplots of global MFA-seq signal of *WT*, *RAD9+/-* and *HUS1+/-* cells, in the indicated phases of cell cycle, at the terminal 0.15 Mb of all chromosomes. In (**B** and **C**), lines represent mean read ratios and the lighter shaded areas indicate SEM.

The online version of this article includes the following figure supplement(s) for figure 7:

**Figure supplement 1.** Extended data for *Figure 7*.

**Figure supplement 2.** MFA-seq signal near chromosome ends upon deficiency of *HUS1* and *RAD9*.

cycle. In either model, the paucity of DNA replication initiation that occurs early in S phase is counterbalanced by DNA replication that can extend beyond conventional S phase.

The view that DNA replication activity is restricted to S phase is being increasingly challenged. Mitotic DNA repair synthesis (MiDAS) has been documented as a means to complete replication of hard-to-duplicate genome features such as common fragile sites and telomeres (*Özer and Hickson, 2018*). Moreover, in the unperturbed cell cycles of both yeast (*Ivanova et al., 2020*; *Torres-Rosell et al., 2007*) and mammalian cells (*Moreno et al., 2016*) onset of mitosis has been documented in the presence of under-replicated genomic regions, leading to DNA synthesis in offspring cells. These findings suggest that a temporal separation between DNA replication and chromosome

segregation does not seem to be a strict rule for eukaryotic cells. However, DNA synthesis outside S-phase is particularly prevalent in cells exhibiting aneuploidy (*Minocherhomji et al., 2015*; *Özer et al., 2018*), suggesting it may be less effective at maintaining genome integrity. Here, we provide evidence that *L. major*, a eukaryotic organism with constitutive mosaic aneuploidy (at least in parasite cells derived from, or proliferating in the insect vector) (*Prieto Barja et al., 2017*; *Dumetz et al., 2017*), achieves full genome duplication using DNA replication that can continue beyond S phase, including G2/M and G1. This finding, in a eukaryotic microbe that is evolutionarily distant from yeast and mammals, suggests that mitosis and cell division prior to complete genome duplication might not necessarily be an inherent feature of genome maintenance and transmission in eukaryotes. In addition, a reliance on DNA replication outside S phase may explain the widespread aneuploidy found in Leishmania. Despite segregation of under-replicated chromosomal regions being mutagenic (*Durkin and Glover, 2007*; *Debatisse et al., 2012*; *Song et al., 2014*), increased cell division rate in these circumstances may compensate for loss of fitness due to mutations. Also, if segregation of under-replicated loci is limited to specific genome compartments, such as the telomeres and subtelomeres, these may serve as a genetic playground, with higher mutation and copy-number variation allowing for increased genetic diversity, as has been proposed in yeast (*Ivanova et al., 2020*; *Brown et al., 2010*). Consistent with this idea, our data suggest that chromosome subtelomeric DNA replication is mainly detectable at the temporal extremes of the conventional Leishmania S phase (either very early in S phase, as seen with the G1 enriched population; or very late in S phase, as seen with the G2/M enriched populations), and such regions have been demonstrated to be particularly prone to copy- number variation (*Bussotti et al., 2018*). Whether post-S phase replication of subtelomeres is widely found in parasites is unclear and identification of molecular cell cycle stages markers will be crucial to determine this. Nonetheless, it is notable that subtelomeric compartments of the *T. brucei* genome share little homology between chromosome homologues, at least in part due to rearrangements among VSG genes (*Müller et al., 2018*). The subtelomeres of *Plasmodium* are also locations of immune response gene diversification, and it may therefore be valuable to dissect the timing of their replication. Indeed, it will be important to ask if core and subtelomere replication timing in any of these parasites, including Leishmania, is influenced by genome organisation in the nucleus (*Müller et al., 2018*; *Bunnik et al., 2019*).

This work also has implications about Leishmania cell cycle checkpoints. To ensure that genome replication is completed before cell division, cells may execute checkpoints to detect ongoing replication forks as cells enter mitosis. If such a checkpoint is operational at all in Leishmania, it must be permissive enough to allow cells to undergo mitosis with considerable ongoing DNA replication. In fact, even if such a cell cycle checkpoint is used, it may not be homogenous across the genome, given the pronounced sensitivity to replication stress at AcH3 sites closer to chromosome ends, contrasting with the mild sensitivity of internal AcH3 sites. This dichotomy may be due to the abundance of repetitive elements at parasite sub-telomeres (*Wickstead et al., 2003*), a known source of replication impediment and genomic instability (*Fouché et al., 2006*; *McMurray, 2010*; *Richard et al., 2008*). Perhaps, in *L. major* the 9-1-1 complex, or its subunits, participates in a specific DNA replication checkpoint responsible for detecting and protecting replication forks stalled by the subtelomeric repeats, allowing DNA synthesis to continue while undergoing mitosis. Such permissiveness would be compatible with the structural and functional diversification of the 9-1-1 subunits (*Damasceno et al., 2016*; *Damasceno et al., 2018*). In this view of the data, DNA replication detected at the subtelomeres may be an extension of replication forks emanating from chromosome-central initiation regions (see below), which are activated in early S phase and are unaffected by HU-induced replication stress. However, it is also possible that the subtelomeric and chromosome-central DNA replication reactions are distinct processes (as discussed below).

Another implication of these findings relates to the coordination of the Leishmania DNA replication process itself. Currently, we cannot establish, among the MFA-seq enriched regions we have mapped, which are *bona fide* origins of replication, recognised by ORC, and which, if any, might correspond with other DNA synthesis events (e.g. MiDAS), which may be ubiquitous throughout the cell cycle. Unlike in *T. brucei* (*Tiengwe et al., 2012a*), binding sites of ORC components have not been mapped in the *L. major* genome, and so it remains to be determined if and how they overlap with the MFA-seq signals we describe. Indeed, despite conservation of predicted ORC components (*Tiengwe et al., 2012b*; *Tiengwe et al., 2014*; *Marques and McCulloch, 2018*; *Marques et al., 2016*), no functional analysis of the initiator has been described in Leishmania (*da Silva et al., 2017*),

meaning we cannot rule out innovations in ORC that could provide S-phase-uncoupled licensing and firing of replication initiation sites, allowing DNA replication outside S phase. Nonetheless, this work shows that the single MFA-seq peak found in each chromosome – the only genomic region clearly activated early in S phase – coincides with KKT1 localised to the boundaries of the transcription units. Here, there is a parallel with *T. brucei*, where the earliest acting origin in each chromosome is bound by ORC and coincides with the mapped centromere (*Tiengwe et al., 2012a*), where at least two KKT factors localise (*Akiyoshi and Gull, 2014*). Moreover, many of the single S-phase MFA-seq peaks in Leishmania are positionally conserved in *T. brucei* (*Marques et al., 2015*), arguing that the putative centromere-focused MFA-seq peaks in Leishmania are ORC-defined origins. In contrast, the subtelomeric regions of intra- and potential post-S-phase DNA replication can only be correlated with AcH3, and we cannot be sure that this form of chromatin is actually a determinant of replication initiation; for instance, DNA replication may emanate from the telomeres. One possibility is that subtelomeric DNA replication is not ORC-derived, but instead derives from DNA repair that is required for the stability of subtelomeric regions, by either controlling copy-number variation in these regions (*Laffitte et al., 2016b*) or by directly driving replication initiation. Such an activity could be related to MiDAS, but may also differ since it appears to be a constitutive component of the Leishmania genome replication programme and looks like it could be related to what has been proposed as a strategy for telomere maintenance (*Bussotti et al., 2018*). The loss of subtelomeric DNA replication in *L. major* cells deficient in RAD9 may provide an insight into the repair machinery involved, since there are parallels with the effects of these mutants and the need for DNA repair activity to direct DNA replication when origins are deleted (*Theis et al., 2010*) or in conditions of replication stress (*Bacal et al., 2018*). Further work will be needed to test if DNA repair, such as homologous recombination, which is known to direct gene amplification (*Laffitte et al., 2016a*), underlies subtelomeric and putative post-S-phase Leishmania DNA replication.

Similar to other eukaryotes, this and previous MFA-seq analysis (*Marques et al., 2015*), as well as SNS-seq mapping (*Lombraña et al., 2016*), have not found association between DNA replication initiation and any specific DNA sequence in *L. major*. Instead, these studies have consistently suggested that features related to transcription activity in *L. major* play a major role in DNA replication initiation. These findings reinforce the idea that Leishmania DNA replication may be initiated in an opportunistic fashion, coupled with transcription initiation and termination events: an association also seen in other eukaryotes (*Lombraña et al., 2013*; *Chen et al., 2019*), although the mechanisms might differ. It is unclear how epigenetic dynamics dictate DNA replication activity in *L. major*. AcH3 seems to correlate with persistent replication initiation throughout the cell cycle. However, when in combination with base J and KKT1, coordinated early S-phase DNA replication initiation is seen, while base J alone is clearly not sufficient for initiation. It should be noted, however, that the ChIP datasets for AcH3, base J and KKT1 lack cell cycle resolution. Thus, it is possible that the order by which each of these modifications is deposited during the cell cycle is a major determinant of the DNA replication initiation outcomes around them. Like in other eukaryotes (*Besnard et al., 2012*; *Comoglio et al., 2015*; *Prorok et al., 2019*), the specific arrangement of G4s around the sites of replication initiation may also be relevant, not only for the kinetics of these forms of chromatin deposition, but also other unknown epigenetic markers and replication-associated factors. Correlation between transcription and GC and AT skews in trypanosomatids has been reported before (*El-Sayed et al., 2005*; *McDonagh et al., 2000*) and is mainly a consequence of constitutive transcription in these organisms. Replication is more restricted in time, and thus only the most efficient replication initiation sites (those around AcH3/base J/KKT1 and, to a lesser extent, AcH3 alone) were locally imprinted with the skew patterns related to DNA replication initiation activity. Distinct GC and AT skews correlate with distinct DNA replication initiation patterns, indicating they may encode some regulatory information themselves, contributing to the space and time orchestration of the DNA replication programme. Alternatively, the skew differences between chromosome-internal and subtelomeric DNA replication initiation sites might have been shaped by the different nature of reactions driving DNA replication from each of these sites.

MFA-seq analysis, as performed previously in Leishmania (*Marques et al., 2015*), is a laborious, time-consuming and expensive technique, requiring purification of cells in different cell cycle stages by flow cytometry, or other techniques such as elutriation. Flow cytometry suggests that in exponentially growing cultures of *L. major*, 25–35% of cells are undergoing genome replication (proportion of cells in S-phase; *Figure 1A*). Thus, we anticipated that the inherent resolution of deep sequencing

should allow us to map DNA replication initiation sites by directly comparing unsorted exponentially growing cells with stationary cells. The success of this approach demonstrates its convenience, which represents a saving in time and cost, which may be particularly useful for comparing conditions or mutants predicted to change DNA replication dynamics. The strategy of genome replication used by Leishmania has been unclear, primarily because of significant discrepancies between previous MFA-seq (*Marques et al., 2015*) and SNS-seq (*Lombraña et al., 2016*) analyses, almost certainly due to differences in sensitivity between the two techniques. While MFA-seq may detect only the most frequent duplication events in the population, SNS-seq is highly sensitive, capable of detecting rarer events. In addition, while SNS-seq has been performed in asynchronous cells, previous MFA-seq used populations enriched for specific cell cycle stages. While the new approach for MFA-seq described here does not fully resolve the discrepancies in predictions of DNA replication activity by these different methodologies (discussed in recent reviews) (*Marques and McCulloch, 2018*; *da Silva et al., 2017*; *Rocha-Granados and Klingbeil, 2016*), the detection of subtelomeric DNA replication provides at least a partial explanation for DNA combing work that has observed two regions of DNA synthesis in a single molecule (*Lombraña et al., 2016*; *Stanojcic et al., 2016*). More importantly, this work suggests that subtelomeric, potentially post-S-phase replication is a core part of genome transmission in Leishmania, and understanding the nature of this form of DNA replication may have implications for understanding genome plasticity in this microbe, and others.

## Materials and methods

### Cell lines and culture

Promastigote forms of *Leishmania major* strain Friedlin were used for MFA-seq data presented in *Figures 1*, *2* and *3*, while *L. major* strain LT252 (MHOM/IR/1983/IR) were used for MFA-seq data presented in *Figures 6* and *7*. Generation of $RAD9^{+/-}$ and $HUS1^{+/-}$ cell lines was previously described (*Damasceno et al., 2013*; *Damasceno et al., 2016*). Cells were cultured at 25°C in HOMEM or M199 medium supplemented with 10% heat inactivated foetal calf serum and 1% penicillin/streptomycin.

### IdU incorporation

Cells were incubated with 150 µM IdU for 30 min and then fixed at −20°C with a mixture (7:3) of ethanol and 1x PBS for at least 16 hr. Next, cells were rinsed with washing buffer (1x PBS supplemented with 1% BSA) and DNA was denatured for 30 min with 2N HCL, followed by neutralisation with phosphate buffer (0.2 M $Na_2HPO_4$, 0.2 M $KH_2PO_4$, pH 7.4). Detection of incorporated IdU was performed with anti-BrdU antibody (diluted in washing buffer supplemented with 0.2% Tween-20) for 1 hr at room temperature. After washing, cells were incubated with anti-mouse secondary antibody conjugated with Alexa Fluor 488 (diluted in washing buffer supplemented with 0.2% Tween-20) for 1 hr at room temperature and then washed. Finally, cells were stained with 1xPBS supplemented with 10 µg.mL$^{-1}$ Propidium Iodide (PI) and 10 µg.mL$^{-1}$ RNAse A and filtered through a 35 µm nylon mesh. FACSCelesta (BD Biosciences) was used for data acquisition and FlowJo software for data analysis. Negative control (omission of anti-BrdU antibody during IdU detection step) was included in each experiment and used to draw gates to discriminate positive and negative events.

### EdU incorporation

Cells were incubated for 1 hr with 10 µM of EdU (Click-iT; Thermo Scientific), then fixed with 3.7% paraformaldehyde for 15 min, adhered into poly-L-lysine coated slides, followed by permeabilisation with 0.5% TritonX100 for 20 min. After washing with PBS supplemented with 3% BSA, cells were subjected to Click-iT reaction, following manufacturers' instructions. DNA was stained with Hoechst 33342. Images were acquired with an SP5 confocal microscope (Leica) and processed with ImageJ software.

### Fluorescent activated cell sorting (FACS) and genomic DNA extraction

FACS and DNA extraction were performed as previously described (*Marques et al., 2015*). Briefly, exponentially growing cells were collected by centrifugation and washed in 1 × PBS supplemented with 5 mM EDTA. Cells were fixed at 4°C in a mixture (7:3) of methanol and 1 × PBS. Prior to sorting,

fixed cells were collected by centrifugation and washed once in 1 × PBS supplemented with 5 mM EDTA, re-suspended in 1 × PBS containing 5 mM EDTA, 10 μg/ml PI and 10 μg/ml RNase A, and passed through a 35 μm nylon mesh. A FACSAria I cell sorter (BD Biosciences) was used to sort cells into lysis buffer (1 M NaCl; 10 mM EDTA; 50 mM Tris–HCl pH 8.0; 0.5% SDS; 0.4 mg/ml proteinase K; 0.8 μg/ml glycogen). Then, sorted cells were incubated for 2 hr at 55℃ and genomic DNA was extracted using Blood and Tissue DNA extraction kit (Qiagen), by omitting the lysis step. Genomic DNA from non-sorted (exponentially growing or stationary) cells was also extracted using Blood and Tissue DNA extraction kit (Qiagen), following the manufacture instructions.

## Whole genome sequencing

Whole genome sequencing of sorted cells used in *Figures 1*, *2* and *3* was previously described (*Marques et al., 2015*). Whole genome sequencing of exponentially growing, stationary and sorted cells (*Figures 4* and *5*) was performed by Eurofins (www.eurofinsgenomics.eu) using Illumina MiSeq paired-end 75 bp or 100 bp sequencing system (Illumina). In order to eliminate differences due to batch effects, each of the early S, late S, G1 and G2 samples per strain/cell line were multiplexed and sequenced in the same run. Sequencing data were uploaded to the Galaxy web platform (usegalaxy.org) for processing (*Afgan et al., 2018*). Quality control was performed with FastQC (http://www.bioinformatics.babraham.ac.uk/projects/fastqc/) and trimomatic (*Bolger et al., 2014*) was used to remove adapter sequences from reads. Reads were mapped to the *L. major Friedlin* reference genome, version 39 (Tritrypdb -http://tritrypdb.org/tritrypdb/), using BWA-mem (*Li and Durbin, 2009*). Reads with a mapping quality score <30 were discarded using SAMtools (*Li et al., 2009*). A summary of filtered reads distribution is shown in *Figure 1—figure supplement 1*.

## Marker frequency analysis (MFA-seq)

After alignment and filtering, reads were compared using the method described previously (*Marques et al., 2015*), with modifications. Briefly, reads were binned in 1.0 kb windows along chromosomes. The number of reads in each bin was used to calculate the ratio between each cell cycle stage (or exponentially growing cells) versus stationary cells, scaled for the total size of the read library. Bins with less than 100 reads in either sample were discarded, as were reads that had a quality score below 30. To reduce noise due to collapsed regions, bins with a ratio above 2.8 were discarded. Also, bins overlapping with other problematic mapping regions, as previously described (*Lombraña et al., 2016*), were also removed. Due to strain differences, further aberrant mapping regions (available upon request) were excluded from samples from LT252 (MHOM/IR/1983/IR). Read depth ratios were normalised as described in *Batrakou et al., 2020*. Due to the prevalence of mosaic aneuploidy in Leishmania, ratio values were also converted into Z scores using in-house R-scripts and the equivalent plots are shown in figure supplements; in either case the data were plotted in 5 kb sliding windows for each individual chromosome, and MFA-seq profiles were represented in a graphical form using Gviz (*Hahne and Ivanek, 2016*). An overview of data analysis is shown in *Figure 1—figure supplement 3*. A comparison between data representation as ratio or Z scores is shown in *Figure 1—figure supplements 4* and *5*.

## Datasets from other studies

ChIP data for AcH3, base J and KKT1 were previously published (*Thomas et al., 2009*; *van Luenen et al., 2012*; *Garcia-Silva et al., 2017*). Experimental mapping of G4-quadruplex in *L. major* was published previously (*Marsico et al., 2019*). Whole genome sequencing used for MFA-seq data presented in *Figures 1*, *2* and *3* was taken from *Marques et al., 2015*.

## GC and AT skew analysis

G and T skews were calculated as (G-C)/(G+C) and (T-A)/(A+T), respectively. Calculations were performed in 1 kb windows using in-house python scripts and subsequently converted into bigwig files. Profiles were represented in a graphical form using Gviz (*Hahne and Ivanek, 2016*).

## Meta-analysis

Unless otherwise stated, all underlying data for metaplots, heatmaps and clustering analyses were generated using deepTools (*Ramírez et al., 2016*), in the Galaxy web platform (usegalaxy. org) (*Afgan et al., 2018*). Metaplots were generated using Prism Graphpad.

## Acknowledgements

We thank all current and previous members of the McCulloch and Tosi labs for input.

## Additional information

### Funding

| Funder | Grant reference number | Author |
|---|---|---|
| Biotechnology and Biological Sciences Research Council | BB/N016165/1 | Luiz RO Tosi<br>Richard McCulloch |
| Biotechnology and Biological Sciences Research Council | BB/R017166/1 | Richard McCulloch |
| European Commission | RECREPEMLE | Jeziel Dener Damasceno |
| Wellcome | 104111 | Richard McCulloch |
| Medical Research Council | MR/S019472/1 | Jeziel Dener Damasceno<br>Richard McCulloch |
| FAPESP | 17/07092-9 | Luiz RO Tosi |

The funders had no role in study design, data collection and interpretation, or the decision to submit the work for publication.

### Author contributions

Jeziel Dener Damasceno, Conceptualization, Resources, Data curation, Formal analysis, Validation, Investigation, Visualization, Methodology, Writing - original draft, Writing - review and editing; Catarina A Marques, Conceptualization, Resources, Formal analysis, Investigation, Methodology, Writing - original draft, Writing - review and editing; Dario Beraldi, Data curation, Software, Formal analysis, Validation, Investigation, Methodology, Writing - review and editing; Kathryn Crouch, Resources, Data curation, Software, Formal analysis, Supervision, Investigation, Methodology, Writing - review and editing; Craig Lapsley, Formal analysis, Validation, Investigation, Writing - review and editing; Ricardo Obonaga, Formal analysis, Investigation, Methodology; Luiz RO Tosi, Conceptualization, Resources, Supervision, Funding acquisition, Investigation, Methodology, Project administration, Writing - review and editing; Richard McCulloch, Conceptualization, Formal analysis, Supervision, Funding acquisition, Investigation, Methodology, Writing - original draft, Project administration, Writing - review and editing

### Author ORCIDs

Jeziel Dener Damasceno (iD) https://orcid.org/0000-0003-2077-3214
Catarina A Marques (iD) http://orcid.org/0000-0003-1324-5448
Kathryn Crouch (iD) http://orcid.org/0000-0001-9310-4762
Richard McCulloch (iD) https://orcid.org/0000-0001-5739-976X

### Decision letter and Author response

Decision letter https://doi.org/10.7554/eLife.58030.sa1
Author response https://doi.org/10.7554/eLife.58030.sa2

## Additional files

### Supplementary files
• Transparent reporting form

### Data availability

Sequences used in this study have been deposited in the European Nucleotide Archive. Data can be accessed using the accession number PRJEB35027.

The following dataset was generated:

| Author(s) | Year | Dataset title | Dataset URL | Database and Identifier |
|---|---|---|---|---|
| Damasceno JD, Marques CA, Beraldi D, Crouch K, Lapsley C, Obonaga R, Tosi LR, McCulloch R | 2020 | Genome duplication in Leishmania major relies on unconventional subtelomeric DNA replication | https://www.ebi.ac.uk/ena/data/view/PRJEB35027 | European Nucleotide Archive, PRJEB35027 |

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
