## [Decision Letter]

**Acceptance summary:**

Previously, by counting marker copy numbers, only a single putative origin of replication had been found for each *Leishmania* chromosome. The problem with this conclusion is that there might not be time to replicate the longer chromosomes. In this paper, the authors confirm that acetylated histone H3 (AcH3), base J and a kinetochore factor co-localise at these previously-mapped loci, which also show G/T skew and G4 patterns. However, they also find indications of additional DNA synthesis that is less strongly cell-cycle regulated, and initiates near the telomeres in regions marked only by AcH3. This may explain how replication of the longer chromosomes can be completed.

**Decision letter after peer review:**

Thank you for submitting your article "Genome duplication in *Leishmaniamajor* relies on unconventional subtelomeric DNA replication" for consideration by *eLife*. Your article has been reviewed by four peer reviewers, including Christine Clayton as the Reviewing Editor and Reviewer #1, and the evaluation has been overseen by Dominique Soldati-Favre as the Senior Editor. The following individual involved in review of your submission has agreed to reveal their identity: Peter Myler (Reviewer #4).

The reviewers have discussed the reviews with one another and the Reviewing Editor has drafted this decision to help you prepare a revised submission.

Summary:

This manuscript includes novel insights into genome replication in *Leishmania*, which suggest various initiation zones are present. The authors performed deep-sequencing analyses of cells sorted by DNA content in different phases of the cell cycle and normalized the read-depth with that obtained from cells stalled at the stationary phase. This unveiled a replication initiation zone close to the telomeres that is more prominent during the G1 and G2 phases of the cell cycle. This initiation zone is altered in replicative stress conditions, what make it distinct to the early S centromere-proximal initiation zone they have previously identified.

Essential revisions:

The reviewers agreed that the results were interesting and that no further experiments are needed, and the *Leishmania* expert said that the manuscript is clearly written, although densely packed at times. However, the two experts on the method agreed that the use of Z-scores was misleading and that you over-interpret the data. It is therefore essential that you follow their recommendations and re-analyse your data without using Z-scores.

Condensed version of their conclusions:

The authors considered two other papers reporting multiple replication initiation sites per chromosome, suggesting that either the SNS technique picks up every initiation site although very infrequent or that the fiber stretching technique is measuring the replication at the kinetoplast circles. However, the MFA is likely measuring replication timing, not replication initiation sites, which would be compatible with the above mentioned works. The alternate possibility is that replication in *Leishmania* starts from multiple inefficient origins that are activated in clusters; one centromere-proximal and another telomere-proximal. The fact that the authors use Z-scores to analyse their MFA data masks non-clustered initiation sites, leading them to interpret their findings of an early initiation zone as an unique early initiation site. Reanalysing the data and moderating their claims will still make their finding of the telomeric-proximal initiation zone interesting for better understanding the biology of these organisms.

Reviewer 3: Lombrana et al., 2016 and Stanojcic et al., 2016, using more precise origin identification methods (Short Nascent Strand analysis and DNA combing) provided strong evidence that origins are much more frequent than this (origin spacing ~180 kb). Although each method of origin mapping has its limitations, marker frequency analysis is relatively imprecise in its ability to map individual origins, and is better suited to analysis of replication timing. It therefore cannot be right for the current paper to ignore the very plausible data obtained by other techniques. This reviewer thus feels that you cannot conclude, from the single MFA-seq peak, that "the majority of DNA replication initiates from a single locus."

Detail from reviewer 2:

1) My main question is about the usage of MFA Z-scores for all the subsequent analysis. I understand that Z-scores are needed when comparing between different strains. Why applying Z-scores when analyzing DNA amounts in cell cycle sorted cells? Normalizing the read depths relative to those of stationary cells would be good enough if the same strain is used; aneuploidies will be corrected and the resolution will be higher, potentially allowing better comparisons with the more sensitive SNS-seq technique, as the authors argue in the Introduction and show in Figure 1—figure supplement 3Ci-ii. Besides, Z-scores magnify relative enrichments what could lead to information loss and to a different interpretation of the data. I would suggest the authors to reanalyze their MFA data without using Z-scores. It might well be that by doing this a different picture will emerge, like DNA replication initiating in two main zones in *Leishmania* chromosomes: one centromere-proximal and another telomere-proximal, but not necessarily from a single origin site in either case.

2) Building on that, looking at the raw data presented in Figure 1—figure supplement 3A it seems that the increased chromosome-central read density is already observed in STA cells (similarly to EXP cells). What happen if STA data are represented as Z-scores in Figure 1—figure supplement 3B? Would the authors interpreted that as DNA replication of some non-stalled cells in the STA population? Or could that be due to other reasons, like amplification of the centromeric regions? Although likely out of the scope of the current work, an important control for all MFA experiments would have been to perform EdU-IP in sorted cells and test if they get similar observations using nascent DNA.

3) Figure 1—figure supplement 3Ci and Cii data suggest to me a replication timing program starting from chromosome ends in G1 and from a centromere-proximal zone in early S, that merge and it is completed in late-S and G2. Could this be related to the transcriptional wave along the cell cycle? If there are not data available on this, at least the authors should comment that possibility in the Discussion. Similarly, such a replication timing scenario could be compatible with a 3D genome architecture in which centromeres and/or telomeres are tethered together in 3D as seen in other unicellular parasites (Bunnik et al., 2019). Replication would initiate from those clustered chromosomal points and then extended to the rest of the genome. This interpretation will be consistent with shorter chromosomes replicating earlier in S-phase (Figure 2). The authors also should discuss this possibility.

4) In conditions of replicative stress or RAD9 deficiency, the authors nicely show that the most telomere-proximal signals are reduced, but more interior ones are not altered or even increased. How do the authors interpret this? These data would fit with the above 3D architecture, where replication origins are activated from telomere-ends even in conditions in which the very distal initiation sites (that might be replicated in a different fashion) are not activated.

[Editors' note: further revisions were suggested prior to acceptance, as described below.]

Thank you for re-submitting your article "Genome duplication in *Leishmaniamajor* relies on unconventional subtelomeric DNA replication" for consideration by *eLife*. Your article has been re-reviewed by two peer reviewers, and the evaluation has been overseen by a Reviewing Editor and Dominique Soldati-Favre as the Senior Editor. The reviewers have opted to remain anonymous.

In conclusion: The data are interesting but neither expert is convinced by your interpretation. It is therefore absolutely essential that you tone down the claims. In particular, please remove the claim that there is a "single origin", neither reviewer thinks that the data support it. The claims about "unconventional replication" or "non-origin directed replication" must also be removed from their title and Abstract. It *is* possible that there is a strong origin region, and multiple less strong origins in other regions – but they are still origins. It is also important to try a bit more to reconcile your data with the fibre data. I've included the reviews verbatim because I think the detailed arguments are important.

Reviewer #2:

The authors have answered my main concern by presenting the MFA data as read-depth rations instead of Z-scores. As they point out, the overall interpretation of the data is not profoundly altered relative to the previous version of the manuscript. On the contrary, the use of read-depth ratios makes clearer that the resolution of the MFA technique is too low to ascribe the mid-chromosomal initiation zone as a single initiation site and, therefore, to further build their claims on the unique G4 patterns and GC-skew at the center of these large early replicating regions as a footprint of replication origins. So, I insist that they should moderate the tone even further on this before publication. Still, I consider the manuscript, and their novel finding of very early subtelomeric replication, of interest for better understanding kinetoplastids biology and support publication in *eLife*. I recommend the authors again to be cautious when interpreting subtelomeric DNA replication as not origin-directed or unconventional. As they also point out in their response to the reviewer's comments, there are no evidences so far on ORC-binding at *Leishmania* chromosomes, so we still don’t know how replication origins are specified in this organism.

Reviewer #3:

I still find the way the results are interpreted quite misleading. Marker Frequency Analysis (MFA) is not a reliable technique for mapping replication origins, though it can reveal the replication timing programme by which different regions of the genome are replicated at different stages of S phase. The results are interpreted as though MFA is giving reliable information about individual origins, and that there is only a single origin driving the bulk of DNA synthesis, which I believe goes against both experimental data (the more reliable fibre analysis) and theoretical considerations of the number of origins required for reliable completion of replication. I would note that base composition bias, although providing some evidence that initiation occurs in a region, is poor evidence that only a single origin is used, and instead is only consistent with most DNA replication being bidirectional around this site.

For example:

“…whereas a single putative origin in each *L. major* chromosome is activated early in S-phase, predicted subtelomeric DNA replication can be detected in all enriched stages of the *L. major* cell cycle.”

“These data are consistent with bidirectional progression of replication forks from a single putative origin in each chromosome”

– “These data may be explained by DNA replication in *L. major* following a programme in which synthesis of a new chromosome initiates at a defined locus in the interior of each chromosome…”

“Since a single major MFA-seq peak was seen in each chromosome in ES cells, the majority of DNA replication is predicted to initiate from a single locus in each case.”

“These data suggest that the simultaneous presence of these three genome factors is a local driver for coordinated DNA replication initiation at a single locus in each chromosome…”

“These data provide clear evidence for base composition bias, consistent with DNA replication initiation, at a single central locus in each of the 36 *L. major* chromosomes.”

“…but their relationship with DNA replication initiation sites mapped by MFA-seq…”

The authors suggest that DNA replication is occurring at cell cycle stages other than S phase, but this runs the danger of being rather circular: how would they define G1 and G2 if not by a lack of DNA synthesis?

The Abstract introduces the concept that telomeric DNA is replicated in a way that is distinct from “origin-directed replication”. I don't understand what this means, and isn't clear to me how this relates to the experimental results and how they are interpreted in the Discussion.

---

## [Author Response]

Essential revisions:The reviewers agreed that the results were interesting and that no further experiments are needed, and the Leishmania expert said that the manuscript is clearly written, although densely packed at times. However, the two experts on the method agreed that the use of Z-scores was misleading and that you over-interpret the data. It is therefore essential that you follow their recommendations and re-analyse your data without using Z-scores.

Every main figure in the paper has been replotted and reanalysed using ratios of read depth in the different cell cycle stages, as requested:

Figure 1D: Z-score plots replaced by read depth ratios.

Figure 2: metaplots and further analysis derived from read-depth ratios of all chromosomes (Figure 1—figure supplement 5A).

Figures 3A, C: plots and metanalysis derived from read-depth ratios.

Figure 4A: Z scores replaced by read depth ratios.

Figure 5B: metaplots of read-depth ratios from all chromosomes. In addition, we have added, as requested, a new panel to each part of the figure showing the regions of transcription termination.

Figure 6: Z-score plots replaced by read depth ratios, and metaplots derived from read-depth ratios of all chromosomes.

Figure 7: Z-score plots replaced by read depth ratios, and metaplots derived from read-depth ratios of all chromosomes.

In plotting read depth ratios, we have normalised the data as described in Batrakou et al., 2020.

Though we understand the reasons behind the reviewers’ caution in using Z-scores to represent the MFAseq data, we hope it is clear that no aspects of the findings (plot profiles, or deeper analyses) are altered when the data were re-plotted or analysed as read depth ratios. To illustrate this point, we have provided direct comparisons of plots and meta-analyses using Z-scores and read depth ratios:

Figure 1—figure supplement 3: MFA-seq profiles in two chromosomes shown as read depth ratios with two different window sizes; Figure 1—figure supplement 3D: MFA-seq profiles in the same chromosomes as Z-scores.

Figure 1—figure supplement 4: a comparison of how data normalisation is used to map read depth ratios for two chromosomes (of differing ploidy) relative to Z-scores.

Figure 1—figure supplement 5: MFA-seq profiles of all chromosomes plotted as normalised read depth ratios (A) or Z-scores (B).

Figure 3—figure supplement 1: MFA-seq data in Figures 2 and 3 (plotted as read depth ratios), plotted and analysed as Z-scores.

Figure 1—figure supplements 2 and 3: further comparisons of plots and meta-analysis as read depth ratios and Z-scores.

We trust that these direct comparisons make clear that the use of Z-scores in the initial submission was not “misleading” and that, rather than us “over-interpret[ing] the data” due to this approach, every aspect of the findings we report are unaltered when the data are represented and analysed as read-depth ratios. Thus, we have made few changes to the text of the manuscript.

We understand that researchers who are relatively unfamiliar with *Leishmania* or related kinetplastids might be sceptical of why the data we present here (and previously) suggest differences from DNA replication dynamics derived from similar analyses in “model” organisms. However, we hope we have reported all data as accurately as possible, and we suggest the emerging picture is one of considerable divergence in kinetoplastid DNA replication dynamics, adding to known divergence in a myriad of core aspects of kinetoplastid cell biology (such as ubiquitous multigene transcription and trans-splicing, and divergence in the ORC and the kinetochore complexes).

Condensed version of their conclusions:The authors considered two other papers reporting multiple replication initiation sites per chromosome, suggesting that either the SNS technique picks up every initiation site although very infrequent or that the fiber stretching technique is measuring the replication at the kinetoplast circles.

We agree that fibre analysis (“DNA combing”) has serious shortcomings in kinetoplastids: in *T. brucei* minicircles and intermediate chromosomes outnumber megabase chromosomes and appear to be segregated by a distinct route; in *Leishmania* episomes can form genome-wide and are replicated in ways that are not understood.

SNS-seq is a much more informative technique to map *Leishmania* DNA replication and we hope we have not given the impression that we discount the important and interesting data reported by Lombraña et al., 2016. However, as expanded on below, SNS-seq predictions of origin number and location in *Leishmania* do not provide a better comparison with origin usage in other eukaryotes than the MFA-seq approach we have adopted.

However, the MFA is likely measuring replication timing, not replication initiation sites, which would be compatible with the above mentioned works.

In *T. brucei* MFA-seq peaks co-localise with ORC binding sites, and both are found at the boundaries of multigene transcription units (Tiengwe et al., 2012). Though we do not have data for ORC binding in *Leishmania* (and are not aware of any reports on this), MFA-seq mapping reveals peaks in the chromosome cores that are also centred on transcription boundaries, ~40% of which are syntenic with *T. brucei*. In addition, the single early S-phase MFA-seq peak in each *Leishmania* chromosome, and the single most pronounced peak in each *T. brucei* chromosome, co-localises with centromeres, which are also found at transcription boundaries. Recent work in *S. pombe* argues that the pre-replication complex is excluded from transcribed genes (Kelly T and AJ Callegari (2019) "Dynamics of DNA replication in a eukaryotic cell" Proc Natl Acad Sci U S A 116(11): 49734982). All MFA-seq data to date in *T. brucei* and *Leishmania* are consistent with a similar exclusion of ORC defined origins from the transcribed regions of the genomes. Of course, we cannot exclude other forms of replication initiation in either parasite, but to date there is no evidence that such initiation is dictated by ORC.

The alternate possibility is that replication in Leishmania starts from multiple inefficient origins that are activated in clusters; one centromere-proximal and another telomere-proximal.

MFA-seq lacks the precision to determine if *Leishmania* DNA replication initiates at a single site or from a cluster of sites. However, clustered initiation sites at the putative origins we map by MFA-seq in the chromosome cores appears incompatible with the highly localised, sharp G/T skew transition we detect only at the single early S-phase MFA-seq peak found in each chromosome. No such clear skew transitions can be detected at the subtelomeres, potentially suggesting a difference in the nature of DNA replication initiation at core and subtelomere locations.

The fact that the authors use Z-scores to analyze their MFA data masks non-clustered initiation sites, leading them to interpret their findings of an early initiation zone as an unique early initiation site. Reanalysing the data and moderating their claims will still make their finding of the telomeric-proximal initiation zone interesting for better understanding the biology of these organisms.

As we have stated above, replotting of the MFA-seq data as ratios rather than Z-scores does not alter any findings. Indeed, the localised G/T-skew patterns at the mapped, core MFA-seq peaks are independent of MFA-seq plotting strategy.

Reviewer 3: Lombrana et al., 2016 and Stanojcic et al., 2016, using more precise origin identification methods (Short Nascent Strand analysis and DNA combing) provided strong evidence that origins are much more frequent than this (origin spacing ~180 kb).

SNS-seq predicts >5000 origins, suggesting an origin spacing closer to ~3 kb, suggesting a much greater density of origins than seen in any other eukaryotes, including where SNS-seq has been used to map initiation (Al Mamun et al., 2016). Origin spacing of ~180 kb is an *extrapolation* from fibre analysis of random DNA fragments, which has limitations in these genomes (as noted above, and by the reviewers).

Although each method of origin mapping has its limitations, marker frequency analysis is relatively imprecise in its ability to map individual origins, and is better suited to analysis of replication timing. It therefore cannot be right for the current paper to ignore the very plausible data obtained by other techniques. This reviewer thus feels that you cannot conclude, from the single MFA-seq peak, that "the majority of DNA replication initiates from a single locus."

As we have stressed above, we really hope we have not given the impression that we discount the interesting SNS-seq data from Lombraña et al., but instead have attempted to understand how the approach reveals a dramatically different prediction for number and location of sites of DNA replication initiation compared with MFA-seq (please see final paragraph of Discussion). The key difference is that SNSseq predicts >5000 initiation sites, most of which are spread across the multigene transcription units, whereas MFA-seq in early S-phase cells maps a single peak in each chromosome at the boundary of the transcription units. Hence, we feel that stating MFA-seq predicts of a single “locus” of DNA replication initiation in each chromosome, irrespective of whether this represents a single initiation site or a tight cluster of sites, is a fair reflection of the dichotomy between the datasets. To reflect this, we have changed “site” to “locus” throughout the manuscript when referring to chromosome-central, putative origins.

Detail from reviewer 2:1) My main question is about the usage of MFA Z-scores for all the subsequent analysis. I understand that Z-scores are needed when comparing between different strains. Why applying Z-scores when analyzing DNA amounts in cell cycle sorted cells? Normalizing the read depths relative to those of stationary cells would be good enough if the same strain is used; aneuploidies will be corrected and the resolution will be higher, potentially allowing better comparisons with the more sensitive SNS-seq technique, as the authors argue in the Introduction and show in Figure 1—figure supplement 3Ci-ii. Besides, Z-scores magnify relative enrichments what could lead to information loss and to a different interpretation of the data. I would suggest the authors to reanalyze their MFA data without using Z-scores. It might well be that by doing this a different picture will emerge, like DNA replication initiating in two main zones in Leishmania chromosomes: one centromere-proximal and another telomere-proximal, but not necessarily from a single origin site in either case.

Please see comments above: we have remapped all the sequence data as normalised read depth ratios, as requested. To reiterate: the findings are the same irrespective of whether the data are represented as ratios or Z-scores; and, crucially, it remains difficult to neatly reconcile MFA-seq and SNS-seq data by either MFA-seq mapping approach.

2) Building on that, looking at the raw data presented in Figure 1—figure supplement 3A it seems that the increased chromosome-central read density is already observed in STA cells (similarly to EXP cells). What happen if STA data are represented as Z-scores in Figure 1—figure supplement 3B? Would the authors interpreted that as DNA replication of some non-stalled cells in the STA population? Or could that be due to other reasons, like amplification of the centromeric regions? Although likely out of the scope of the current work, an important control for all MFA experiments would have been to perform EdU-IP in sorted cells and test if they get similar observations using nascent DNA.

When read coverage from STA cells is represented as Z scores we are also able to see enrichment around the putative early S phase origins. The best explanation would be the presence of ~4% of cells in the population still undergoing some sort of DNA synthesis (see Figure 1B).

The centromeres in *Leishmania* are currently only defined by KKT1 binding; no functional analysis has shown that the single KKT1 binding site in each chromosome acts as a centromere, and etoposide mapping (which identified centromeres in *T. brucei*) has not been reported. Thus, we currently have limited information to build on for the centromeres and further work is warranted (in the future).

3) Figure 1—figure supplement 3Ci and Cii data suggest to me a replication timing program starting from chromosome ends in G1 and from a centromere-proximal zone in early S, that merge and it is completed in late-S and G2. Could this be related to the transcriptional wave along the cell cycle? If there are not data available on this, at least the authors should comment that possibility in the Discussion. Similarly, such a replication timing scenario could be compatible with a 3D genome architecture in which centromeres and/or telomeres are tethered together in 3D as seen in other unicellular parasites (Bunnik et al., 2019). Replication would initiate from those clustered chromosomal points and then extended to the rest of the genome. This interpretation will be consistent with shorter chromosomes replicating earlier in S-phase (Figure 2). The authors also should discuss this possibility.

No work has examined 3D genome organisation in *Leishmania*, unlike in *T. brucei* (Muller et al., 2018). In fact, in *T. brucei*, correlation of 3D structure and replication timing is hampered by the very repetitive and extensive subtelomeres, which have confounded attempts at MFA-seq mapping. Nonetheless, this is a good suggestion and we have added the following text to the Discussion:

“Indeed, it will be important to ask if core and subtelomere replication timing in any of these parasites, including Leishmania, is influenced by genome organization in the nucleus”.

4) In conditions of replicative stress or RAD9 deficiency, the authors nicely show that the most telomere-proximal signals are reduced, but more interior ones are not altered or even increased. How do the authors interpret this? These data would fit with the above 3D architecture, where replication origins are activated from telomere-ends even in conditions in which the very distal initiation sites (that might be replicated in a different fashion) are not activated.

We have argued that these data might indicate distinct forms of DNA replication, but the reviewers’ suggestion of an influence of 3D architecture is fair: please see text addition above.

[Editors' note: further revisions were suggested prior to acceptance, as described below.]

In conclusion: The data are interesting but neither expert is convinced by your interpretation. It is therefore absolutely essential that you tone down the claims. In particular, please remove the claim that there is a "single origin", neither reviewer thinks that the data support it. The claims about "unconventional replication" or "non-origin directed replication" must also be removed from their title and Abstract. It is possible that there is a strong origin region, and multiple less strong origins in other regions – but they are still origins. It is also important to try a bit more to reconcile your data with the fibre data. I've included the reviews verbatim because I think the detailed arguments are important.

We have gone through the manuscript and removed all instances where we have used the term “single origin” (see detailed responses to reviewer #3) and, in addition, have changed the title and Abstract as follows:

Title: Genome duplication in *Leishmania major* relies on persistent subtelomeric DNA replication

Abstract: “DNA replication is needed to duplicate a cell’s genome in S-phase and segregate it during cell division. Previous work in *Leishmania* detected DNA replication initiation at just a single region in each chromosome, an organisation predicted to be insufficient for complete genome duplication within S-phase. Here, we show that acetylated histone H3 (AcH3), base J and a kinetochore factor co-localise in each chromosome at only a single locus, which corresponds with the previously mapped DNA replication initiation regions and is demarcated by localised G/T skew and G4 patterns. In addition, we describe previously undetected subtelomeric DNA replication in G2/M and G1 phase-enriched cells. Finally, we show that subtelomeric DNA replication, unlike chromosome-internal DNA replication, is sensitive to hydroxyurea and dependent on 9-1-1 activity. These findings indicate that *Leishmania*’s genome duplication programme employs subtelomeric DNA replication initiation, possibly extending beyond S-phase, to support predominantly chromosome-internal DNA replication initiation within S-phase.”

We have discussed, as far as possible, ongoing differences between our data and fibre analysis (an approach that is limited, as it has not been used with reference to genome structure or content): see response to reviewer #3.

Reviewer #2:The authors have answered my main concern by presenting the MFA data as read-depth rations instead of Z-scores. As they point out, the overall interpretation of the data is not profoundly altered relative to the previous version of the manuscript. On the contrary, the use of read-depth ratios makes clearer that the resolution of the MFA technique is too low to ascribe the mid-chromosomal initiation zone as a single initiation site and, therefore, to further build their claims on the unique G4 patterns and GC-skew at the center of these large early replicating regions as a footprint of replication origins. So, I insist that they should moderate the tone even further on this before publication. Still, I consider the manuscript, and their novel finding of very early subtelomeric replication, of interest for better understanding kinetoplastids biology and support publication in eLife. I recommend the authors again to be cautious when interpreting subtelomeric DNA replication as not origin-directed or unconventional. As they also point out in their response to the reviewer's comments, there are no evidences so far on ORC-binding at Leishmania chromosomes, so we still don´t know how replication origins are specified in this organism.

We entirely accept the major limitation in interpreting our data: current lack of mapping of *Leishmania* ORC to the genome (unlike in *T. brucei*, where ORC binding and MFA-seq peaks colocalise at the ends of multigene transcription units: Tiengwe et al., 2012). We have stressed this in the Discussion:

“Currently, we cannot establish, among the MFA-seq enriched regions we have mapped, which are bona fide origins of replication, recognised by ORC, and which, if any, might correspond with other DNA synthesis events (e.g. MiDAS), which may be ubiquitous throughout the cell cycle”.

To ensure this limitation is made yet more explicit, we have altered the text throughout to remove inappropriate references to “origins” and, instead, to refer to MFA-seq predicted DNA replication initiation “regions”. Please see responses to reviewer #3 for detailed changes.

As regards subtelomeric DNA replication, we have limited discussion of potential modes of replication to the Discussion and have altered the Abstract to remove the suggestion that this reaction is origin independent:

“we show that subtelomeric DNA replication, unlike chromosome-internal DNA replication, is sensitive to hydroxyurea and dependent on 9-1-1 activity”.

In addition, the term “unconventional” has been replaced with “persistent” in the title.

Reviewer #3:I still find the way the results are interpreted quite misleading. Marker Frequency Analysis (MFA) is not a reliable technique for mapping replication origins, though it can reveal the replication timing programme by which different regions of the genome are replicated at different stages of S phase. The results are interpreted as though MFA is giving reliable information about individual origins, and that there is only a single origin driving the bulk of DNA synthesis, which I believe goes against both experimental data (the more reliable fibre analysis) and theoretical considerations of the number of origins required for reliable completion of replication. I would note that base composition bias, although providing some evidence that initiation occurs in a region, is poor evidence that only a single origin is used, and instead is only consistent with most DNA replication being bidirectional around this site.For example:“…whereas a single putative origin in each L. major chromosome is activated early in S-phase, predicted subtelomeric DNA replication can be detected in all enriched stages of the L. major cell cycle.”“These data are consistent with bidirectional progression of replication forks from a single putative origin in each chromosome”“These data may be explained by DNA replication in L. major following a programme in which synthesis of a new chromosome initiates at a defined locus in the interior of each chromosome…”“Since a single major MFA-seq peak was seen in each chromosome in ES cells, the majority of DNA replication is predicted to initiate from a single locus in each case.”“These data suggest that the simultaneous presence of these three genome factors is a local driver for coordinated DNA replication initiation at a single locus in each chromosome…”“These data provide clear evidence for base composition bias, consistent with DNA replication initiation, at a single central locus in each of the 36 L. major chromosomes.”“…but their relationship with DNA replication initiation sites mapped by MFA-seq…”

As stated above (reviewer #2), we entirely accept the two consequences of the unavailability of ORC mapping in *Leishmania*: we cannot say if MFA-seq peaks represent ORC-defined origins, and nor can we say if the peaks represent initiation at a discrete site or over wider stretches of sequence. Thus, to address the reviewer’s legitimate concerns, we have made a large number of changes throughout the manuscript to explicitly highlight the limitations of the approach and available data by removing all inappropriate use of the term “origin” and replacing it with DNA replication initiation “region”.

Below we detail the most important changes, including in response to the reviewer’s requests:

Introduction:

“However, unlike all previously characterised eukaryotes, mapping of DNA replication using Marker Frequency Analysis coupled with deep sequencing (MFA-seq) detected only a single clear region of replication initiation in each chromosome of *Leishmania,* a grouping of single-celled parasites. If these MFA-seq regions represent origins (as they do in *S. cerevisiae*), these data would suggest a DNA replication programme in *Leishmania* that is unprecedented in eukaryotes and, indeed, contrasts with the multiple origins mapped in the chromosomes of *Trypanosoma brucei ,* a kinetoplastid relative of *Leishmania* (see below)”.

“Consistent with previous MFA-seq mapping, DNA replication initiation in early S-phase could only be clearly detected from a single internal region in each chromosome”.

“Thus, we reveal that two potentially distinct forms of DNA synthesis make up the genome replication programme of *L. major*, with a single predominant S-phase region of initiation in each chromosome supplemented by subtelomeric DNA synthesis activity that appears to extend beyond S-phase.”

Results section:

“Taken together, these data indicate two things. First, sites of subtelomeric DNA synthesis, predicted by change in sequence copy number, appear spatially separate from previously mapped, predominantly chromosome-central regions of DNA replication initiation and can be found in virtually every *L. major* chromosome. Second, whereas a single replication initiation region in each *L. major* chromosome is activated early in S-phase, predicted subtelomeric DNA replication initiation can be detected in all enriched stages of the *L. major* cell cycle.”

“These data are consistent with bidirectional progression of replication forks emanating from a single predominant region in each chromosome”

“These data may be explained by DNA replication in *L. major* following a programme in which synthesis of a new chromosome initiates at a single predominant region in the interior of each chromosome in early S-phase and progresses at a consistent rate towards the chromosome ends, but with DNA replication continuing even as cells navigate from LS through G2/M until they enter G1”.

“Since a single major MFA-seq peak was seen in each chromosome in ES cells, the majority of DNA replication is predicted to initiate from a single region in each case.”

“These data suggest that the simultaneous presence of these three genome factors is a local driver for coordinated DNA replication initiation at a single region in each chromosome in ES but does not promote DNA replication initiation in other cell cycle stages.”

“These data provide clear evidence for base composition bias, consistent with DNA replication initiation at a central region in each of the 36 *L. major* chromosomes.”

“G4s have been predicted to co-localise with SNS-seq signal in *Leishmania*, but their relationship with DNA replication initiation regions mapped by MFA-seq has not been explored.”

Discussion section:

“Our data show that genome duplication predominantly starts from a single region in every chromosome in early S phase”.

“In either model, the paucity of DNA replication initiation that occurs early in S phase is counterbalanced by DNA replication that can extend beyond conventional S phase.”

“In this view of the data, DNA replication detected at the subtelomeres may be an extension of replication forks emanating from chromosome-central initiation regions (see below), which are activated in early S phase and are unaffected by HU-induced replication stress.”

“Moreover, many of the S-phase MFA-seq peaks in *Leishmania* are positionally conserved in *T. brucei* , arguing that the putative centromere-focused MFA-seq peaks in *Leishmania* are ORC-defined origins.”

In the Discussion we acknowledge the differences in DNA replication mapping predicted by MFA-seq, SNSseq and fibre analyses (the latter of which was noted by previous reviewers to be potentially unreliable in kinetoplastids due to abundant episomes and minichromosomes):

“While the new approach for MFA-seq described here does not fully resolve the discrepancies in predictions of DNA replication activity by these different methodologies (discussed in recent reviews), the detection of subtelomeric DNA replication provides at least a partial explanation for DNA combing work that has observed two regions of DNA synthesis in a single molecule.”

The authors suggest that DNA replication is occurring at cell cycle stages other than S phase, but this runs the danger of being rather circular: how would they define G1 and G2 if not by a lack of DNA synthesis?

This suggestion is consistent with recent studies where it has been demonstrated that in both yeast (Ivanova et al., 2020, Nat Commun. 11(1):2267, and Torres-Rosell et al., 2007) and mammalian cells (Moreno et al., 2016) there is DNA synthesis outside conventional S phase.

Flow cytometry analysis that we show of *Leishmania* cells’ DNA content reveals a canonical cell cycle profile, which allowed us to define and isolate G1 (2C DNA content), S (2-4C) and G2/M (4C) cells, arguing that most DNA replication is cell cycle regulated.

Nonetheless, we have been cautious when discussing these issues:

Abstract: “These findings indicate that *Leishmania*’s genome duplication programme employs subtelomeric DNA replication initiation, possibly extending beyond S-phase, to support predominantly chromosome-internal DNA replication initiation within S-phase.”

Introduction: “Thus, we reveal that two potentially distinct forms of DNA synthesis make up the genome replication programme of *L. major*, with a single predominant S-phase region of initiation in each chromosome supplemented by subtelomeric DNA synthesis activity that appears to extend beyond Sphase.”

Discussion: “Consistent with this idea, our data suggest that chromosome subtelomeric DNA replication is mainly detectable at the temporal extremes of the conventional *Leishmania* S phase (either very early in S phase, as seen with the G1 enriched population; or very late in S phase, as seen with the G2/M enriched populations), and such regions have been demonstrated to be particularly prone to copy number variation”.

The Abstract introduces the concept that telomeric DNA is replicated in a way that is distinct from “origin-directed replication”. I don't understand what this means, and isn't clear to me how this relates to the experimental results and how they are interpreted in the Discussion.

We have removed mention of “origin-directed replication” from the Abstract, which now states: “we show that subtelomeric DNA replication, unlike chromosome-internal DNA replication, is sensitive to hydroxyurea and dependent on 9-1-1 activity”.

The results presented in the subsection “DNA replication initiation around chromosome ends is more susceptible to replication stress”, and in the subsection “Replication initiation at chromosome ends requires RAD9 and HUS1”, provide evidence that replication initiation from chromosomeinternal regions might rely on distinct machineries from replication initiation at the subtelomeres.

Nonetheless, we have also toned down these conclusions:

“Thus, we reveal that two potentially distinct forms of DNA synthesis make up the genome replication programme of *L. major*, with a single predominant S-phase region of initiation in each chromosome supplemented by subtelomeric DNA synthesis activity that appears to extend beyond Sphase.”